# CS2W: A Chinese Spoken-to-Written Style Conversion Dataset with Multiple Conversion Types

**Zishan Guo, Linhao Yu, Minghui Xu, Renren Jin, Deyi Xiong**[*]

College of Intelligence and Computing, Tianjin University, Tianjin, China

{guozishan,linhaoyu,xuminghui,rrjin,dyxiong}@tju.edu.cn

## Abstract

Spoken texts (either manual or automatic transcriptions from automatic speech recognition (ASR)) often contain disfluencies and grammatical errors, which pose tremendous challenges to downstream tasks. Converting spoken into written language is hence desirable. Unfortunately, the availability of datasets for this is limited. To address this issue, we present CS2W, a Chinese Spoken-to-Written style conversion dataset comprising 7,237 spoken sentences extracted from transcribed conversational texts. Four types of conversion problems are covered in CS2W: disfluencies, grammatical errors, ASR transcription errors, and colloquial words. Our annotation convention, data, and code are publicly available at https://github.com/guozishan/CS2W.

## 1 Introduction

Automatic speech recognition (ASR) plays a vital role in a wide range of NLP application scenarios, such as simultaneous interpretation, where verbal utterances are transcribed into spoken style texts. These transcriptions serve as fundamental inputs to plenty of downstream tasks. However, they often inherently contain disfluencies, grammatical errors, and colloquial words, which pose tremendous challenges on downstream tasks. Automatically correcting errors and editing spoken into written language would significantly benefit downstream tasks that are usually trained on canonical texts. Developing such spoken-to-written style conversion models usually requires labeled data that cover different phenomena in ASR-transcribed spoken style texts.

Unfortunately, existing datasets usually focus on a single type of spoken style, such as disfluencies. Consequently, models trained on these datasets cannot address all spoken style issues.

| Category | Example |
|---|---|
| Disfluency | 嗯，你早上那个吃饭了吗？ 
 Uh, did you eat that this morning? 
 你早上吃饭了吗？ 
 Did you eat this morning? |
| ASR Transcription Errors | 他会踢橡胶球。 
 He can kick a rubber ball. 
 他会踢香蕉球。 
 He can perform a banana kick.[1] |
| Grammatical Errors | 我看过这部电。 
 I have seen this electric.[2] 
 我看过这部电影。 
 I have seen this movie. |
| Colloquial Words | 比赛中他一直划水。[3] 
 He kept paddling in the game. 
 比赛中他一直偷懒。 
 He has been lazy in the game. |

Table 1: Examples of the four conversion types. In each category, the first pair of sentences is the original spoken style text and its machine translation output while the second is the normalized text converted from the original spoken style text and its machine translation output.

To bridge this gap, we propose CS2W, a large-scale fine-grained Chinese Mandarin spoken-to-written style conversion dataset, developed on the Real Spontaneous Dialogue Speech dataset MagicData-RAMC (Yang et al., 2022). CS2W consists of 7,237 annotated instances, covering four major conversion problems corresponding to the majority of spoken phenomena. For each conversion instance, we manually annotate the spans and types of the corresponding conversion problems.

We conduct a thorough and in-depth analysis on Chinese spoken texts and summarize four conversion problems: disfluencies, ASR transcription errors, grammatical errors, and colloquial words. The

---

[*]Corresponding author

[1]In Chinese, "rubber ball" is pronounced the same as "banana kick".

[2]In Chinese, the word "电影" (movie) is pronounced as "dian ying," and the individual character "电" (electric) is pronounced as "dian." The error in this sentence is that only half of the word "电影" is mentioned.

[3]In Chinese, the colloquial word "划水" refers to the act of slacking off during work or study.

four conversion problems are common in Chinese spoken texts and cover the major tasks in spoken-to-written style conversion (i.e., grammatical and ASR error correction, and simplification).

- **Disfluency**: Repetitions, restarts, or repairs in spontaneous communication.

- **ASR transcription errors**: Occasional homophone mistakes in ASR transcriptions.

- **Grammatical errors**: Missing words, incorrect syntax structures, etc., similar to those occurring in written style texts.

- **Colloquial words**: Problems related to colloquial words that differ from written language.

Table 1 shows examples of the four types of conversion problems and their impact on machine translation.

In comparison to existing datasets that focus on grammatical errors or disfluencies, our dataset contains more conversion types and closely aligns with the distribution of real-world spoken data. For example, in the commonly used SWITCHBOARD corpus (Godfrey et al., 1992), instances usually contain a few disfluent words (Charniak and Johnson, 2001), and more than half of those disfluencies consist of repetitions (Shriberg, 1996). In contrast, our dataset frequently contains various types of grammatical errors and disfluencies within a single sentence.

Our contributions are as follows:

1. We curate and release CS2W, the first open-source Chinese dataset for spoken-to-written style conversion. The dataset is derived from real-world spontaneous conversations. We provide fine-grained annotation along with written style manually normalized texts. Additionally, we establish a comprehensive set of criteria for spoken-to-written style conversion classification and annotation.

2. We conduct an in-depth analysis on the distribution of spoken-to-written style conversion problems and identify new types of disfluencies.

3. We conduct benchmark evaluation experiments on CS2W to evaluate the performance of cutting-edge large language models on

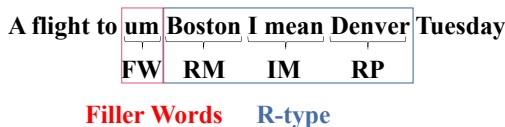

Figure 1: A sentence with disfluencies annotated in the style of Shriberg (1996) and the SWITCHBOARD corpus (Godfrey et al., 1992). FW=Filler Words, RM=Reparandum, IM=Interregnum, RP=Repair.

spoken-to-written language conversion. Experiment results demonstrate that the conversion from spoken to written language effectively improves the performance of downstream tasks.

## 2 Related Work

Previous studies have treated disfluency detection and grammatical error correction as separate tasks. We hence review their progress separately. A comprehensive comparison of GEC, disfluency detection, and spoken-to-written style conversion datasets is presented in Table 2.

**Disfluency Detection and Related Datasets** Disfluency is a common phenomenon in spoken language and is generally divided into two categories: R-types and Filler words. Filler words have no meaning and are often used to indicate pauses and hesitations of the speaker. They are enumerable and easy to detect, for example, in English, Filler words include "uh", "you know", "well" and so on. R-types include repeat, restart, and repair. As shown in Figure 1, a repair type disfluency includes a reparandum ("Boston") and an interregnum ("I mean"), followed by its repair ("Denver "). A repetition type has the same reparandum and interregnum. A restart type has only a reparandum without interregnum, meaning the speaker starts a new topic. The most prevalent approach to disfluency correction involves treating it as a sequence tagging task, aiming to produce fluent discourse by identifying and removing disfluent segments (Zayats et al., 2016). Traditional models of disfluency detection use syntactic features (Honnibal and Johnson, 2014), language models (Johnson et al., 2004), or rhyme-based learning features (Zayats and Ostendorf, 2019), while more recent disfluency detection models mainly make use of pre-trained neural representations (Jamshid Lou et al., 2018). Most of these models rely on manually annotated data.

Unfortunately, few disfluent detection datasets

| Dataset | Task | Type of Conversion Covered | Annotation Paradigm | #Sentences | Domain | Language |
|---|---|---|---|---|---|---|
| FCE | GEC | GEC | Error-coded | 34K | Essay | EN |
| AESW | GEC | GEC / S2W | Error-coded | 1.2M | Journal Articles | EN |
| JFLEG | GEC | GEC | Direct Rewriting | 1.5K | TOFEL Exam | EN |
| WI-LOCNESS | GEC | GEC | Direct Rewriting | 3.7K | Language-learning Website | EN |
| NLPCC18-task | GEC | GEC | Direct Rewriting | 717K | Language-learning Website | CH |
| CGED | GEC | GEC | Error-coded | 7.2K | HSK Exam | CH |
| SWITCHBOARD | DC | DC | Error-coded | 265K | Telephone Conversation | EN |
| PhoDisfluency | DC | DC | Error-coded | 5.8K | Telephone Conversation | VI |
| Japanese S2W | S2W | GEC / DC / S2W | Direct Rewriting | 52K | Telephone Conversation | JA |
| CS2W | S2W | GEC / DC / S2W | Direct Rewriting Error-coded | 7.2K | Telephone Conversation | CH |

Table 2: Comparison between our dataset and other datasets. DC: Disfluency Correction. GEC: Grammatical Error Correction. S2W: Spoken-to-Written Style Conversion.

are publicly available. The SWITCHBOARD corpus (Godfrey et al., 1992) consists of transcribed telephone conversations between two individuals discussing a specific topic, which include disfluencies. Compared to this dataset, CS2W contains a higher density of disfluent sentences.

Due to the sparsity of disfluency data, Dao et al. (2022) construct a Vietnamese disfluency dataset PhoDisfluency by manually introducing disfluency perturbations to the fluent sentences. In contrast, CS2W is collected from natural, spontaneous conversations, and is the first open-source Chinese dataset containing disfluency issues.

**Grammatical Error Correction and Related Datasets** Common errors in Chinese grammatical error correction (GEC) datasets include spelling errors, missing words, redundant words, incorrect word order, collocation errors, etc. Seq2Seq models, based on RNN/CNN or Transformer, are usually used for Chinese GEC tasks. Hinson et al. (2020) first propose a Seq2Edit model for Chinese GEC, which achieves comparable performance with the Seq2Seq counterparts. Most Seq2Edit-based models use PLMs like BERT (Devlin et al., 2019) to initialize their encoders. Li and Shi (2021) apply a non-autoregressive neural machine translation model to Chinese GEC.

Numerous datasets have been proposed for GEC. These datasets primarily use two main paradigms to label the data: *error-coded* and *direct rewriting*. In the error-coded paradigm, annotators are tasked with explicitly identifying erroneous spans in the original sentence, specifying the error type, and subsequently making corrections. For instance,

FCE (Vougiouklis et al., 2018) is an early large-scale English GEC dataset using the error-coded paradigm, comprising raw text produced by English learners who complete their First Certificate in English. Another example is AESW (Daudaravicius et al., 2016), sourced from a professional editing company, which focuses on assessing the level of technical writing and includes questions on not only writing style conversion but grammatical errors. Additionally, the NLPCC18-task (Zhao et al., 2018) dataset originates from Lang-8[4], a language learning website where native speakers can freely select essays by learners for revision. Conversely, the direct rewriting paradigm instructs annotators to rephrase the input sentence directly, providing a corresponding grammatically correct version without altering the original meaning. An example is JFLEG (Napoles et al., 2017), developed with reference to the TOEFL exam. It places emphasis on preserving the overall fluency of the rewritten text. Moreover, WI-LOCNESS (Bryant et al., 2019) encompasses two distinct datasets, derived from essays written by students with English as either their first or second language. Finally, CGED (Rao et al., 2018, Rao et al., 2020) is designed to diagnose grammatical errors in Chinese, based on the HSK (Hanyu Shuiping Kaoshi, Test of Chinese Level) examination.

In contrast to the existing GEC datasets, our CS2W dataset places a stronger emphasis on grammatical errors occurring in the spoken domain rather than the written domain. Notably, while all previously established datasets feature a single

[4]https://lang-8.com/

annotation paradigm, the CS2W dataset stands out by offering two distinct annotation paradigms.

**Spoken-to-Written Style Conversion and Related Datasets** Spoken-to-written style conversion can be formulated as a monolingual translation task (Wubben et al., 2010), mapping from a spoken style text to a normalized written style text. A variety of approaches have been proposed to address these monolingual translation challenges, including noisy channel models and hidden Markov models (Johnson and Charniak, 2004; Ferguson et al., 2015; Matusov et al., 2006) used earlier. In recent years, neural sequence conversion models have demonstrated superior performance (See et al., 2017). Nevertheless, these models require a sizable parallel corpus for training since they learn the relationship between input and output sequences directly in an end-to-end manner. Therefore, it is essential to curate a large-scale parallel corpus of spoken and written language.

Publicly available spoken-to-written style conversion data are scarce, with the majority of datasets being private. Ihori et al. (2020) curate the Parallel Corpus for Japanese Spoken-to-Written Style Conversion, which annotates not only grammatical errors but also special conversion types in Japanese. These types include restoring postpositional particle expressions, hiragana symbols, etc. CS2W is a spoken-written style conversion dataset, where the conversion problems are categorized according to Chinese linguistic characteristics.

## 3 Dataset Curation

In this section, we elaborate on the creation of the CS2W dataset.

### 3.1 Data Source

CS2W is built upon the existing MagicData-RAMC dataset (Yang et al., 2022), which consists of 351 sets of spontaneous conversations in Chinese Mandarin. Each set features natural conversations between two speakers on a single topic, and it includes audio files and transcribed texts that retain real-world disfluencies, grammatical errors, and ASR transcription errors. We manually select sentences from the ASR transcriptions, which are self-contained in meaning but with conversion problems for annotation. In total, we collect 7,237 sentences for further annotation. A comprehensive description of the data extraction process can be found in Appendix A.

### 3.2 Annotation Guidelines

The spoken-to-written style conversion normally involves lexical and syntactical editing and style transfer. The former is similar to grammatical error correction, dealing with disfluencies, ASR transcription errors, and grammatical errors while the latter is for the translation of colloquial words into canonicalized words with the same meaning used in written style texts.

To annotate the selected sentences, we employ two established annotation paradigms: error-coded and direct rewriting, commonly used in grammatical error correction datasets (Zhao et al., 2018; Rao et al., 2020), which have been described in Section 2.

Initially, for each sentence, we use the error-coded paradigm to meticulously identify spoken-to-written style conversion problems and determine the types of these problems (i.e., $\in$ *disfluency*, *ASR transcription error*, *grammatical error*, *colloquial word*). We also pinpoint the specific spans within the sentences where these problems manifest. It's essential to recognize that a sentence may exhibit multiple conversion problems simultaneously.

However, it is worth noting that (Sakaguchi et al., 2016) highlight certain challenges associated with the error-coded paradigm, particularly concerning consensus among annotators regarding the identification of spans and problem types. Furthermore, complex annotation paradigms sometimes lead annotators to neglect the aspect of sentence fluency, resulting in unnatural expressions. Therefore, we also incorporate the direct rewriting paradigm into our annotation. This involves rewriting the spoken style language directly into a written form, producing references that are not only grammatically correct but also fluent and in a written style. Additionally, the labeled conversion problems in the error-coded paradigm need to be consistent with those corrected in the written style reference.

Table 3 provides a breakdown of the labeling for different conversion types. To enhance annotation consistency among different annotators, we develop comprehensive guidelines in Appendix B, which offer detailed descriptions of our annotation convention along with illustrative annotation examples.

### 3.3 Annotation Process

To ensure the consistency and quality of our annotations, we implement a two-round annotation

| MainType | Sub-Type | SouceSentence | Reference |
|---|---|---|---|
| Disfuency | R-Type | 有些应用这个有些应用需要付费。
Some apps, uh, some apps require payment. | 有些应用需要付费。
Some apps require payment. |
| | Filler Words | 他年纪大了有什么头疼的毛病。
He has, I mean, headaches as he gets older. | 他年纪大了有头疼的毛病。
He has headaches as he gets older. |
| ASR Transcription Errors | - | 他会踢橡胶球。
He can kick a rubber ball. | 他会踢香蕉球。
He can perform a banana kick. |
| Grammatical Errors | Missing Words | 你知道他什么了吗?
Do you know what he? | 你知道他说什么了吗?
Do you know what he said? |
| | Redundant Words | 它们的皮毛很有光泽,
可以用肉眼很难看出来。
Their fur is shiny and can be
hardly seen with the naked eye. | 它们的皮毛很有光泽,
可以用肉眼看出来。
Their fur is shiny and can be
seen with the naked eye. |
| | Incorrect Word Order | 昨天看了新买的一部电影我在电视上。
Yesterday watched a newly
purchased movie I on TV. | 昨天我在电视上看了新买的一部电影。
Yesterday I watched a
newly purchased movie on TV. |
| Colloquial Words | - | 这明明是你的功劳,却被同事抢
走了,你真是一个大怨种。[5]
This is obviously your credit,
but your coworkers took it away.
You're such an unlucky guy. | 这明明是你的功劳,却被同事抢
走了,你真是太倒霉了。
This is obviously your credit,
but your coworkers took it away.
You're so unlucky. |

Table 3: Example source sentences and references for the four main types and subtypes of spoken-to-written style conversion. Underlined phrases are spans with conversion problems.

process.

In the first round of annotation, we enlist the expertise of eight Chinese native speakers as part-time annotators after pre-annotation training. We require that each conversion problem within a given spoken style text be annotated in accordance with established conventions, assuring the validity and uniformity of the annotations.

In the second round, we conduct a manual evaluation and re-annotation, with the authors of this paper serving as senior annotators. Each annotated instance is scrutinized according to the answers to the following three core questions:

1. Are the type and span annotations correct? This step ensures the accuracy of annotations under the error-coded paradigm.

2. Does the written style reference faithfully retain the meaning of the original spoken style text? This step ensures the accuracy of annotations under the direct rewriting paradigm.

3. Is the modification in the written language reference consistent with the conversion problems being labeled? This ensures the alignment between the annotations under the error-coded paradigm and those under the direct rewriting paradigm.

If the annotators in the second round encounter any inconsistencies with the first-round annotations, they submit their annotations to the senior annotators. The senior annotators then perform a comparative analysis of the results from both rounds and issue the final annotation verdict. Detailed guidelines for resolving inconsistencies are provided in Appendix B.

## 4 Dataset Analysis

We provide data statistics and analyses on conversion type distribution, sentences with multiple conversion problems, and a new subtype of disfluency existing in our dataset, which is absent in previous studies.

### 4.1 Overall Statistics

Table 4 presents the overall statistics of our dataset. We randomly shuffle all the annotated texts and partition them into training, development, and test sets with a proportion of 8:1:1. The training set consists of 5,789 texts, while both the development and test sets contain 724 texts each. Not surprisingly, most erroneous texts contain only one error. After Chinese word segmentation via the PKUseg tool (Luo et al., 2019), we obtain a total of 143,738 tokens. On average, each text contains approximately 12.82 tokens. Additionally, the average length of each span is 2.02, and each sentence contains an average of 1.35 spans.

---

[5]In Chinese, "大怨种" is an Internet phrase used to describe people who are aggrieved but have no way to complain.

| | Train | Dev | Test | All |
|---|---|---|---|---|
| #Sentences | 5,789 | 724 | 724 | 7,237 |
| w/ 1 error | 4,211 | 596 | 597 | 5,404 |
| w/ 2 errors | 1,069 | 107 | 112 | 1,288 |
| w/ 3 errors | 317 | 15 | 11 | 343 |
| w/ ≥4 errors | 192 | 6 | 4 | 202 |
| #Spans | 8,022 | 889 | 881 | 9,792 |
| Avg. #spans | 1.386 | 1.228 | 1.217 | 1.353 |
| Avg. spans_len | 2.003 | 1.883 | 2.069 | 2.020 |
| #Characters | 121,907 | 10,844 | 10,987 | 143,738 |
| Avg. #characters | 21.058 | 15.033 | 15.175 | 19.862 |
| #Tokens | 78,549 | 7,120 | 7,116 | 92,785 |
| Avg. #tokens | 13.569 | 9.834 | 9.829 | 12.821 |

Table 4: Data statistics of CS2W. "w/n" means that a sentence contains n errors.

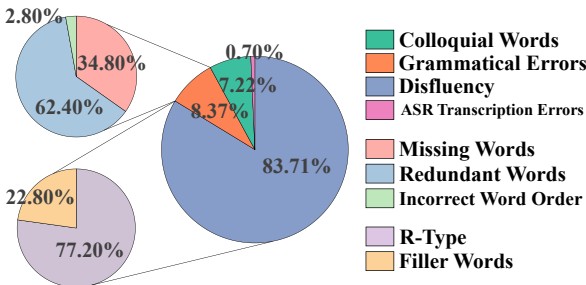

Figure 2: Distributions of conversion types and their subtypes.

## 4.2 Conversion Type Distribution

To assess the distribution over different conversion types, we calculate the proportions of each conversion type present in the dataset. Results are illustrated in Figure 2. Since sentences in our dataset often contain multiple conversion types, the percentage of each conversion type is obtained by dividing the number of occurrences of the conversion type by the total number of conversions in the source sentence. As shown in Figure 2, disfluency dominates in conversion problems, and notably, the percentage of R-type conversion problems, which is often challenging to rectify, is substantially high. Grammatical errors constitute a relatively minor portion, accounting for 6.9% of the dataset. Within this category, the principal error types include word missing and word redundancy.

## 4.3 Analysis on Sentences with Multiple Conversion Problems

In contrast to previous disfluency detection and grammatical error correction datasets, where each sentence typically has only one problem, as depicted in Figure 3, our CS2W dataset presents a notable departure. Specifically, we find that 25.34% of the single sentences within CS2W exhibit multiple conversion problems, with a significant 31.86% of these sentences displaying more than two types

of conversion problems. This observation underscores the prevalence of disfluency problems, as a substantial number of sentences with other types of conversion problems also include disfluencies. Surprisingly, over 50% of sentences featuring grammatical errors or colloquial words also include disfluency problems. This suggests a likely connection between disfluencies and the emergence of grammatical errors and colloquial words, as disfluencies often arise from pauses in thought during spontaneous language production, potentially contributing to these issues.

## 4.4 A New sub-type of Disfluency

Our analysis of the most prevalent conversion problem, disfluency, in the context of spoken-to-written style conversion, has revealed a novel disfluency pattern. In the conventional R-type disfluency, the reparandum (the incorrect portion) typically precedes the repair (the corrected portion). This pattern aligns with the common observation that speakers often correct themselves upon realizing an error. However, in this distinct disfluency pattern, the repair precedes the reparandum, as illustrated in Figure 4. In other words, an utterance fragment is originally said correctly but immediately said incorrectly when it is repeated.

Traditional disfluency correction models commonly delete the portion preceding the disfluency to generate a grammatically correct sentence. However, when confronted with this specific pattern, such an approach could potentially compromise the accuracy of the correction process.

## 5 Experiment

We conducted experiments on the curated dataset to evaluate the performance of the advanced open-source large language models (LLMs) on Chinese spoken-to-written style conversion.

### 5.1 Dataset

We ensured randomness in our data selection process by shuffling all the annotated texts. Subsequently, we partitioned these texts into training, development, and test sets, maintaining a distribution ratio of 8:1:1. The training set encompasses 5,789 texts, while both the development and test sets consist of 724 texts each.

### 5.2 Baselines

We used the following baseline models:

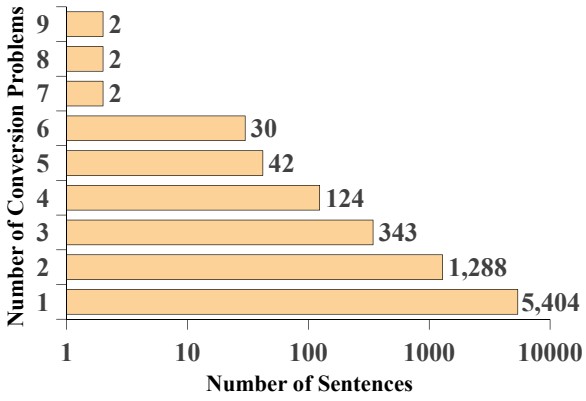

(a) Statistics on the number of conversion problems contained in a single sentence.

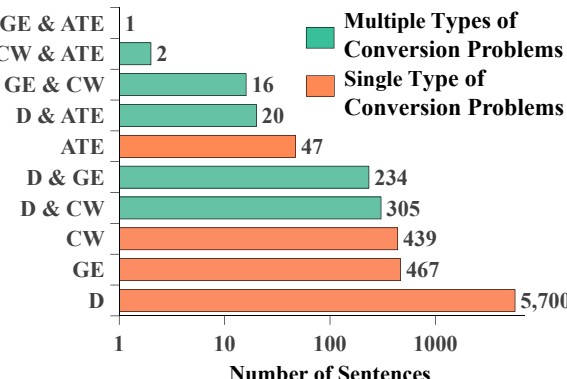

(b) Statistics on the types of conversion problems contained in a single sentence.

Figure 3: Statistics on the number and type of conversion problems contained in a single sentence. The horizontal axis uses logarithmic coordinates. D: Disfluency. ATE: ASR Transcription Errors. GE: Grammatical Errors. CW: Colloquial Words.

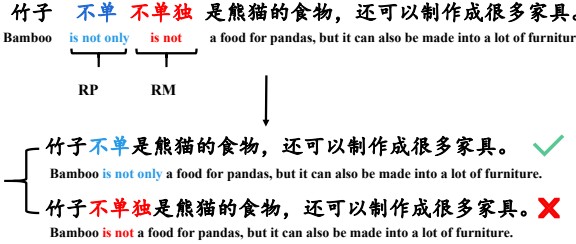

Figure 4: A new type disfluency example, RM=Reparandum, RP=Repair; Generally, RM is before RP.

- BART (Lewis et al., 2020): an encoder-decoder language model where a bidirectional encoder is used to encode inputs and a left-to-right decoder is used to generate outputs. We used the Chinese BART[6] proposed by Shao et al. (2021).

- CPT (Shao et al., 2021): a language model that shares knowledge between natural language understanding (NLU) and natural language generation (NLG) to boost performance. We used CPT-base[7] and CPT-large.[8]

- BLOOM (Scao et al., 2022): a decoder-only transformer language model that is trained on the ROOTS corpus (Laurençon et al., 2022), which contains a huge amount of data for 46 natural languages and 13 programming languages.

- BELLE (Yunjie Ji, 2023): a model based on BLOOM and finetuned with Chinese data combined with 50,000 pieces of English data, resulting in good Chinese instruction following and response generation capabilities.

- ChatGLM[9]: an open bilingual language model based on General Language Model (GLM) (Du et al., 2022) framework, trained on 1T tokens of Chinese and English corpus, supplemented by supervised fine-tuning, feedback bootstrap, and reinforcement learning with human feedback.

- GPT3.5-turbo[10]: a model trained on diverse data sources, which is an advanced language model based on the GPT-3.5 architecture.

We fine-tuned BART and CPT with the training set of CS2W and tested BLOOM, BELLE, ChatGLM, and GPT3.5-turbo under the zero- and 5-shot settings. The demonstrations used for the 5-shot setting and the method of selecting them are described in detail in Appendix C. The numbers of parameters of the models, as well as the hyperparameters, are described in detail in Appendix D.

## 5.3 Metrics

We evaluated models with BLEU (Papineni et al., 2002) and ROUGE-L (Lin, 2004). Referring to most grammatical error correction tasks, we also used the word-based MaxMatch scorer to calculate

---

[6]https://huggingface.co/fnlp/bart-base-chinese
[7]https://huggingface.co/fnlp/cpt-base
[8]https://huggingface.co/fnlp/cpt-large

[9]https://github.com/THUDM/ChatGLM-6B
[10]https://openai.com/product

| Model | Setting | $F_{0.5}$ | P | R | B-1 | B-2 | B-3 | B-4 | Rouge-L |
|---|---|---|---|---|---|---|---|---|---|
| BART-base | fine-tuning | 0.3049 | 0.3521 | 0.2641 | 0.1391 | 0.0434 | 0.0113 | 0.0020 | 0.4259 |
| BART-large | | 0.4486 | 0.5025 | 0.3351 | 0.3051 | 0.1476 | 0.0696 | 0.0323 | 0.5129 |
| CPT-base | | 0.8421 | 0.8503 | 0.8255 | 0.7348 | 0.6598 | 0.6026 | 0.5470 | 0.8539 |
| CPT-large | | **0.8469** | **0.8526** | **0.8377** | **0.7376** | **0.6669** | **0.6117** | **0.5577** | **0.8589** |
| BLOOM-7B | 0-shot | 0.4400 | 0.4097 | 0.7140 | 0.2783 | 0.2203 | 0.1808 | 0.1441 | 0.4319 |
| BELLE-7B-0.2M | | 0.5583 | 0.5537 | 0.6013 | 0.4390 | 0.3076 | 0.2249 | 0.1631 | 0.5215 |
| BELLE-7B-2M | | 0.6262 | 0.6232 | 0.6651 | 0.4829 | 0.3574 | 0.2731 | 0.2040 | 0.5735 |
| ChatGLM-6B | | 0.6414 | 0.6295 | 0.7227 | 0.5765 | 0.4741 | **0.4134** | **0.3627** | 0.6892 |
| GPT3.5-turbo | | **0.7285** | **0.7260** | **0.7673** | **0.5846** | **0.4802** | 0.4042 | 0.3373 | **0.7023** |
| BLOOM-7B | 5-shot | 0.4520 | 0.4180 | 0.7480 | 0.2982 | 0.2409 | 0.2000 | 0.1609 | 0.4601 |
| BELLE-7B-0.2M | | 0.5376 | 0.5289 | 0.6276 | 0.2903 | 0.2186 | 0.1733 | 0.1347 | 0.5135 |
| BELLE-7B-2M | | 0.6308 | 0.6282 | 0.6662 | 0.4885 | 0.3519 | 0.2790 | 0.2167 | 0.5860 |
| ChatGLM-6B | | 0.6648 | 0.6593 | 0.7326 | 0.5924 | 0.4872 | 0.4261 | 0.3785 | 0.6943 |
| GPT3.5-turbo | | **0.7778** | **0.7777** | **0.7954** | **0.6882** | **0.5862** | **0.5087** | **0.4393** | **0.7466** |

Table 5: Automatic evaluation results for models on the CS2W dataset.

P/R/F values. The method for calculating these values is described in Appendix E.

## 5.4 Main Results

Table 5 presents the performance of benchmark models on CS2W. Notably, the fine-tuned CPT-large delivers outstanding results across all metrics. Importantly, it significantly outperforms the advanced LLMs under both the zero-shot and few-shot (5-shot) settings. This underscores the continued significance of CS2W in the context of Chinese Spoken-to-Written Style Conversion, even in the era of LLMs.

Under the zero-shot setting, ChatGLM exhibits a slight advantage over GPT3.5-turbo in terms of BLEU-3 and BLEU-4 scores, while GPT3.5-turbo excels in the remaining metrics. In contrast, under the 5-shot setting, GPT3.5-turbo emerges as the frontrunner, achieving the highest scores across all metrics. It's worth noting that, except for BELLE-7B-0.2M, all models demonstrate improved performance under the 5-shot setting when compared to their zero-shot setting performance.

## 5.5 Performance on Four Conversion Types

Table 6 presents the benchmark model's performance across various types of conversions. The results from human evaluation are based on the accuracy of the first round of annotation. The results show that CPT-large and GPT3.5-turbo performed similarly across different conversion types. They achieve the highest accuracy in correcting disfluencies while encountering more challenges in addressing colloquial words, which aligns with the distribution of problems observed in CS2W.

| Type | CPT-large | GPT3.5-turbo | Human |
|---|---|---|---|
| Disfluency | 87.99 | 80.53 | 75.72 |
| ASR Transcription Errors | 77.77 | 71.21 | 90.47 |
| Grammatical Errors | 83.59 | 80.73 | 61.54 |
| Colloquial Words | 57.96 | 54.58 | 86.73 |
| Mixed Type | 74.21 | 62.93 | 84.12 |

Table 6: $F_{0.5}$ scores for CPT-large and GPT3.5-turbo for different conversion types.

## 5.6 Contribution of Spoken-to-Written Language Conversion to Downstream Tasks

To assess the impact of spoken-to-written language conversion on downstream tasks, we employed Chinese-to-English machine translation as a representative task. We randomly selected 100 normalized text references from the test set and then manually translated them into English to serve as references for the Chinese-to-English machine translation task.

We utilized OPUS-MT[11] (Tiedemann and Thottingal, 2020) to translate the source sentences of these 100 references, along with the corresponding outputs of CPT-base and CPT-large. This approach allowed us to quantitatively evaluate the effect of spoken-to-written style conversion on the Chinese-to-English machine translation task. The results, presented in Table 7, underscore the substantial positive impact of spoken-to-written language conversion as a preprocessing step for spoken language.

---

[11] https://huggingface.co/Helsinki-NLP/opus-mt-zh-en

|            | BLEU-1 | BLEU-2 | BLEU-4 |
|------------|--------|--------|--------|
| Source sentence | 0.587 | 0.511 | 0.410 |
| CPT-base | 0.711 | 0.635 | 0.524 |
| CPT-large | 0.727 | 0.645 | 0.536 |

Table 7: BLEU-1, BLEU-2, and BLEU-4 of the source sentence and the corresponding outputs of CPT-base and CPT-large on the Chinese-to-English translation task.

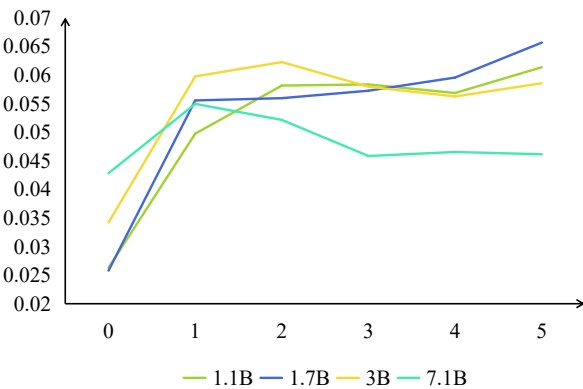

Figure 5: BLEU-2 scores for BLOOM series models with different numbers of demonstrations.

Notably, the BLEU-4 score shows an improvement of 0.126 after the conversion done by CPT-large.

### 5.7 Results with different numbers of demonstrations provided to LLMs

We conducted further experiments on the BLOOM series models to investigate the impact of the number of demonstrations in the prompts on these LLMs. Specifically, we selected BLOOM series models with parameter sizes of 1.1B[12], 1.7B[13], 3B[14], and 7.1B[15] and performed experiments across a range of demonstration numbers, from 0 to 5. The outcomes of these experiments are graphically represented in Figure 5. Most models achieve significant improvement under the 1-shot setting, but performance drops and then increases with more demonstrations. We conjecture that this is due to model sensitivity to prompts. From zero to one demonstration, the model performance improves significantly due to the relevant prompts. In experiments with two to three demonstrations, limited prompt diversity leads to a performance decrease. With more demonstrations and diversity, the model performance gradually increases again.

---

[12]https://huggingface.co/bigscience/bloom-1b1
[13]https://huggingface.co/bigscience/bloom-1b7
[14]https://huggingface.co/bigscience/bloom-3b
[15]https://huggingface.co/bigscience/bloom-7b1

## 6 Conclusion

In this paper, we have presented a dataset CS2W, which is the first open-source Chinese spoken-to-written conversion dataset. The dataset covers four types of conversion problems commonly occurring in Chinese spoken texts. We manually annotate the type and span for each conversion problem and provide high-quality written style normalized texts. The dataset is used as a benchmark testbed to evaluate the performance of advanced LLMs on spoken-to-written style conversion and would promote future research on this underexplored direction.

### Limitations

For all zero-shot and few-shot experiments, we used the same prompt for all models. However, prompt selection is important for large language models. We plan to use more prompts and prompt engineering methods to conduct experiments on the curated dataset in the future.

### Ethics Statement

All data used in this study are freely available to the public. The raw data are from a public dataset built in previous work. We follow the policy of using these data without infringing any copyright issues. The curated dataset in this study is for academic research purposes only. All annotators are well paid according to the number of their annotations.

### Acknowledgements

The present research was supported by the Key Research and Development Program of Yunnan Province (No. 202203AA080004) and Zhejiang Lab (No. 2022KH0AB01). We would like to thank the anonymous reviewers for their insightful comments.

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

## A Data Extraction

MagicData-RAMC is an ASR transcription dataset that comprises 351 sets of spontaneous conversations in Chinese Mandarin. Each set features natural dialogues between two speakers discussing a single topic, and it includes both audio files and transcribed texts that preserve disfluencies, grammatical errors, and ASR transcription errors. We manually select sentences from the ASR transcriptions, ensuring they are self-contained in meaning but with conversion problems for annotation.

In transcribed text, we do not extract the following three types of sentences: Incomplete Sentences, Sentences that require context, and Sentences that are too short.

### A.1 Incomplete sentences

In spontaneous conversations, a speaker often breaks off abruptly or is interrupted by another speaker, resulting in many incomplete sentences in transcriptions.

e.g. 我和他刚刚准备出去玩，就。(He and I were just getting ready to hang out but.)

This indicates that the speaker stops speaking at the word "but" or is interrupted by someone else. Incomplete sentences cannot be easily understood by annotators and, as a result, may not be accurately annotated.

### A.2 Sentences that require context

Spontaneous conversations have continuity and many sentences need to be understood in context.

e.g. Speaker 1: 你这次数学考试考了多少分？(Speaker 1: What grade did you get on this math test? )

Speaker 2: 一百多一点，比不上你。(Speaker 2: little over a hundred, no more than you. )

Speaker 2's speech is a response to Speaker 1's question, which may not be fully comprehensible without considering Speaker 1's speech. Sentences that require context can also be challenging to annotate accurately.

### A.3 Sentences that are too short

In spoken language, phrases such as "no problem," "yes," and "okay" are frequently employed, and these are considered too general. Therefore, we refrain from selecting sentences with fewer than 5 tokens.

| First Level | Second Level |
|---|---|
| Disfluency | R-type |
| | Filler words |
| ASR Transcription Errors | - |
| Grammatical Errors | Missing Words |
| | Redundant Words |
| | Incorrect Word Order |
| Colloquial Words | - |

Table 8: Two-level classification system of conversion problem types.

In conclusion, we select complete sentences with conversion problems whose lengths are appropriate.

## B Annotation Guidelines

### B.1 Task Statement

Automatic speech recognition (ASR) plays a vital role in a wide range of NLP application scenarios. Spoken language, which serves as a fundamental input for plenty of downstream tasks, is transcribed into written text in a spoken style. However, they often inherently contain disfluencies, grammatical errors, and colloquial words. This dataset consists of transcribed texts with conversion problems for annotation. For each sentence, the annotator needs to annotate the type and the range of conversion problems and write the corresponding written language.

### B.2 Conversion Type Definitions and Examples

We further categorize the conversion problem with a two-level classification system shown in Figure 8. Each conversion problem is described in detail next.

#### B.2.1 Disfluency

The elements that make a sentence not fluent are referred to as "disfluency", which can be categorized into R-type and Filler Words based on their structures.

**Filler Words** Filler Words, such as "uh" and "ah," have no specific meaning and are often used to indicate pauses and hesitations in the speaker's discourse. Additionally, common words like "yeah" and "okay" are also sometimes classified as filler words.

e.g. 好吧，这是个，嗯，一个不错的主意。 (Well, this is, you know, a good plan.)

In this sentence, the phrases "well" and "you know" lack specific meaning and should be annotated as "Filler words."

**R-type** The standard structure of R-type disfluency encompasses three elements: the reparandum, an optional interregnum, and the associated repair. The reparandum consists of words that the speaker initially intends to discard, representing an unintended inclusion in the utterance. This section typically comprises one or more words slated for repetition or correction. The interregnum, often comprising fixed phrases like "uh" or "you know," serves as a non-lexicalized component, contributing filler words without specific meaning. Lastly, the repair phase involves correcting or repeating words from the reparandum, thereby refining the overall coherence of the utterance.

e.g. 让我们，我的意思是，让我来解决这个问题 。 (Let us, I mean, let me work on the problem.)

In the provided example, the sentence "Let us" functions as the reparandum, embodying the words originally unintended for inclusion. The subsequent phrase "let me" constitutes the repair, correcting the preceding reparandum. The interregnum in this instance is "I mean," a non-essential filler phrase devoid of substantive meaning.

### B.2.2 ASR Transcription Errors

ASR Transcription Errors are occasional homophone mistakes in ASR transcriptions. CS2W is built upon the existing MagicData-RAMC dataset (Yang et al., 2022), which comprises 351 sets of spontaneous conversation speech in Chinese Mandarin and their ASR transcriptions. Consequently, there are occasional homophone mistakes in some sentences.

e.g. 这个艺术家很有菜花。 (The artist is very cauliflower.[16])

According to the intended meaning of the sentence, the correct version should be "The artist is very talented." Therefore, "cauliflower" needs to be annotated as an "ASR Transcription Error."

### B.2.3 Grammatical Errors

The transcription text of spoken language often includes grammatical errors because speakers in

---

[16]In Chinese, "cauliflower" and "talented" have the same pronunciation.

---

conversations often lack careful thinking. Common grammatical errors in spoken language include Missing Words, Redundant Words, and Incorrect Word Order.

**Missing Words** Missing Words include missing subjects, missing predicates, missing objects, missing function words, and missing modifiers.

e.g. 那时我们有机会扳平比分，但是我们没有机会。 (We had a chance to equalize, but we didn't it.)

This sentence is missing a verb. The correct sentence is "We had a chance to equalize, but we didn't take it".

**Redundant Words** Redundant Words include redundant subjects, redundant predicates, redundant objects, redundant function words, and redundant modifiers.

e.g. 它们的皮毛很有光泽,可以用肉眼很难看出来。 (Their fur is shiny and can be hardly seen with the naked eye.)

The modifier in this sentence is redundant. The fur is shiny so it should be visible to the naked eye. The word "hardly" should be deleted.

**Incorrect Word Order** Incorrect Word Order is also common in spoken transcribed texts because of the frequent inversions in spoken language.

e.g. 昨天看了新买的一部电影我在电视上。 (Yesterday watched a newly purchased movie I on TV.)

In Chinese, the correct sentence is "Yesterday I watched a newly purchased movie on TV."

### B.2.4 Colloquial Words

Spoken language often contains informal expressions, such as some popular Internet phrases, which are called "Colloquial Words."

e.g. 这明明是你的功劳，却被同事抢走了，你真是一个大怨种 。 (This is obviously your credit, but your coworkers took it away. You're such an unlucky guy.)

We need to replace all informal expressions with formal ones. In Chinese, the Internet phrase "大怨种" is used to describe people who are aggrieved but have no way to complain. Therefore, we should replace it with the more formal expression "You're unlucky."

### B.2.5 Mixed Type

In real spontaneous conversations, a single sentence often contains multiple conversion problems.

e.g. 在国内成立野牛，这个，水牛研究中心，有利于帮助适应人工环境。 (The establish-

ment of the bison, I mean, buffalo research center in the country will help to adapt to an artificial environment.)

This sentence contains both disfluency and grammatical errors. The corrected version should be: "The establishment of the buffalo research center in the country will help them adapt to an artificial environment."

## B.3 Annotation Rules

We built an annotation platform to accelerate our annotation progress. When using it, the annotator needs to select the conversion type and annotation range of the current sentence. Then, the annotator provides the written language corresponding to this spoken language. Next, we will present the annotation rules for different conversions. Please note that on the annotation platform, each sentence is word-segmented into individual words, each of which can be selected to make it easier for the annotator to annotate the conversion range.

**Disfluency** The annotator selects the Disfluency button on the annotation platform. For Filler Words, the annotator should annotate their range. For R-type, both the reparandum and the interregnum should be annotated, but the repair does not need to be annotated. This is because the sentence can be corrected by simply deleting the reparandum and the interregnum.

**ASR Transcription Errors** The annotator selects the ASR Transcription Errors button on the annotation platform and annotates the range of the ASR Transcription Errors.

**Grammatical Errors** The annotator selects the Grammatical Errors button on the annotation platform. For Missing Words, the annotator needs to annotate the two words before and after the missing part. For Redundant Words, the annotator needs to annotate the redundant part. As for Incorrect Word Order, the entire sentence has to be annotated.

**Colloquial Words** The annotator selects the Colloquial Words button on the annotation platform and annotates the range of the colloquial words.

**Mixed Type** First, the annotator selects the button for the initial conversion type and annotates the range of that conversion. Subsequently, the annotator sequentially selects the buttons for the other conversion types and annotates their respective ranges.

## B.4 Inconsistent Treatment

To ensure annotation consistency and quality, we implement a two-round annotation process. In the first round, we enlist eight native Chinese speakers as part-time annotators after providing pre-annotation training. In the second round, we conduct manual evaluation and re-annotation, with the authors of this paper serving as senior annotators. In the event of inter-annotator disagreements, the annotator in the second round reannotates the sentence and submits the results of both rounds to the senior annotator. There are two scenarios of inter-annotator disagreements.

The first scenario is when the annotator from either the first or second round makes an incorrect annotation.

e.g. 当他回到车车间时，已经有了明显的变化。 (When he returned to the ga-, garage, had changed markedly.)

The sentence exhibits two conversion problems, namely Disfluency and Missing Words. The annotator in the first round accurately annotates the disfluency but overlooks the grammatical errors. In the second round, the annotator correctly annotates both conversion problems. The senior annotator will then determine the correct annotation.

The second case is that the sentence can be corrected in multiple ways, which is common in grammatical errors.

e.g. Source sentence: 如果人们连续看上四五个小时的电视节目，就会感到十分疲劳。 (If people watch TV programs for four or five hours in a row, will feel very tired.)

Target sentence 1: 人们如果连续看上四五个小时的电视节目，就会感到十分疲劳。 (People in case of watching TV programs for four or five hours in a row will feel very tired.)

Target sentence 2: 如果人们连续看上四五个小时的电视节目，他们就会感到十分疲劳。 (If people watch TV programs for four or five hours in a row, they will feel very tired.)

This sentence lacks a subject and has two potential solutions. First, considering the conversion as an "Incorrect Word Order," the corresponding written language is target sentence 1. Second, considering the conversion as a "Missing Word," the written language is target sentence 2. In this scenario, the senior annotator selects the solution with a smaller edit distance. If the edit distances are equal, the senior annotator opts for the first-round solution.

## C Prompt Templates

### C.1 Instructions Used in Main Results

Regarding the prompts used for zero- and few-shot settings, we tried two different prompts on BLOOM-7B under the zero-shot and 5-shot settings.

**Prompt 1**: "下面有一个口语到书面语风格转换任务,请把口语修改为书面语: 口语: {源句} 书面语:" (Here's a spoken-to-written style conversion task, please rewrite the spoken language into the written language: spoken: {source sentence} written:).

**Prompt 2**: "下面有一个语法纠错任务,请把错误的文本修改为正确的文本: 错误文本: {源句} 正确文本: " (Here's a grammatical error correction task, please correct the wrong text into the right text: wrong text: {source sentence} correct text: )

Prompt 2 outperforms Prompt 1 on all metrics. We speculate that the model may struggle to comprehend the definition of the spoken-to-written language conversion task. However, grammatical error correction is a widely-used task, and the four conversion problems, except for colloquial words, can be considered either simple or complex grammatical errors, allowing it to perform well. Hence, we adopt Prompt 2 for the rest of our experiments.

### C.2 Demonstrations Used in Main Results

Under the zero-shot setting, the prompt is "下面有一个语法纠错任务，请把错误的文本修改为正确的文本：错误文本：{源句} 正确文本：" (Here's a grammatical error correction task, please correct the wrong text into the right text: wrong text: {source sentence} correct text:).

We expect LLMs to be capable of correcting all types of conversion problems. Therefore, it is crucial to ensure the diversity of demonstrations provided to LLMs. Given that CS2W encompasses four types of conversion problems, we incrementally add demonstrations of different types as the number of demonstrations increases. The demonstrations for each conversion problem are randomly selected from the validation set. These demonstrations differ from the input sentences in the test set and contain only one conversion problem. Additionally, since CS2W is dominated by disfluency as the primary conversion problem, we specifically select two demonstrations, one for R-type and another for Filler Words, under the 5-shot setting.

In Section 5.4 and 5.7, the demonstrations used in 5-shot experiments are shown in Figure 11.

### C.3 Impact of Demonstrations Diversity

While our initial intuition suggested that including a higher diversity of conversion types in demonstrations would enable LLMs to address more types of conversions and subsequently improve results, our findings have led us to reconsider this notion. To test this hypothesis, we randomly selected five demonstrations from the validation set and repeated the selection process three times, resulting in three distinct prompts used for the 5-shot setting. The specific demonstrations included in each prompt are detailed in Table 12. Notably, Prompt 1 incorporates demonstrations with three different conversion types, offering the highest diversity, while Prompt 2 exclusively includes demonstrations featuring disfluency problems, resulting in the lowest diversity.

We tested BELLE-7B-2M, ChatGLM-6B, and GPT3.5-turbo under the 5-shot settings with these three prompts. The results, in comparison to those presented in Section 5.4, are summarized in Table 13.

In summary, our experimental findings challenge our initial hypothesis that greater diversity in examples would consistently enhance the ability of LLMs to address a wider range of conversion types and lead to improved results. Surprisingly, it is not always the case. The prompt with the most diversity, Prompt 1, displays the weakest performance. In contrast, Prompt 2 and Prompt 3, which feature fewer types of conversions but a higher concentration of demonstrations with disfluencies, delivered more favorable results.

We attribute this phenomenon to the prevalence of disfluency within the CS2W dataset. When the number of demonstrations with disfluencies surpasses a certain threshold, the overall performance tends to improve. However, it's worth noting that the dataset's distribution of conversion types may play a pivotal role in these results. In a scenario where the four conversion types were more balanced, prompts with greater diversity might have exhibited improved performance.

## D Model Parameter Sizes and Hyperparameters

The numbers of parameters of the models used in the experiments are shown in Table 9.

| Model | Parameter Count |
|-------|-----------------|
| BART-base | 139M |
| BART-large | 406M |
| CPT-base | 121M |
| CPT-large | 393M |
| BLOOM | 7B |
| BELLE | 7B |
| ChatGLM | 6B |

Table 9: The number of parameters of the models used in the experiments.

| Model | Decoding Temperature |
|-------|----------------------|
| BLOOM-7B | 1.0 |
| BELLE-7B-0.2M | 0.35 |
| BELLE-7B-2M | 1.0 |
| ChatGLM-6B | 0.95 |
| GPT3.5-turbo | 0.7 |

Table 10: The decoding temperature of LLMs used in the experiments.

The decoding temperature of LLMs used in the experiments is shown in Table 10.

## E   P/R/F Calculation Method

Firstly, the gold sentence and the models' outputs are word-segmented using the PKUNLP word segmentation (WS) tool (Luo et al., 2019), and then we calculate the number of maximal matches according to the WS results. The P, R, $F_{0.5}$ measure the different rates between the set of the model's output edits $e_1, ..., e_2$ and the set of gold sentence edits $g_1, ..., g_2$ for all sentences:

$$P = \frac{\sum_{i=1}^{n} |e_i \cap g_i|}{\sum_{i=1}^{n} |e_i|}$$
$$R = \frac{\sum_{i=1}^{n} |e_i \cap g_i|}{\sum_{i=1}^{n} |g_i|}$$
$$F_{0.5} = \frac{1.25 * P * R}{0.25 * P + R}$$

Where we define the intersection between $e_i$ and $g_i$ as:

$$e_i \cap p_i = \{e \in e_i | \exists g \in g_i (match(e, g))\}$$

| Number | Demonstration (Wrong Text) | Demonstration (Correct Text) | Conversion Type |
|---|---|---|---|
| 1 | 我不不太喜欢听那种歌曲。
I don't don't like this type of music. | 我不太喜欢听那种歌曲。
I don't like this type of music. | Disfluency R-type |
| 2 | 你呃喝酒喝的比较多了。
You, uh, have been drinking a lot. | 你喝酒喝的比较多了。
You have been drinking a lot. | Disfluency Filler words |
| 3 | 我爸会糖醋丸子。
My dad can sweet and sour dumplings. | 我爸会炸糖醋丸子。
My dad can make sweet and sour dumplings. | Grammatical Errors |
| 4 | 我想给游戏氪金。
I want to spend money on the game. | 我想给游戏充钱。
I want to recharge in the game. | Colloquial Words |
| 5 | 这件事情我无可奉报。
I have nothing to report in this matter | 这件事情我无可奉告。
I have nothing to say in this matter. | ASR Transcription Errors |

Table 11: Five demonstrations used in main experiments.

| Number | Demonstration (Wrong Text) | Demonstration (Correct Text) | Conversion Type |
|---|---|---|---|
| Prompt 1 Demonstration 1 | 我为什么要套路别人？
Why do I try to trap other people? | 我为什么要哄骗别人？
Why do I try to lie to other people? | Colloquial Word |
| Prompt 1 Demonstration 2 | 他他让我假装睡觉。
He he asked me to pretend to sleep. | 他让我假装睡觉。
He asked me to pretend to sleep. | Disfluency |
| Prompt 1 Demonstration 3 | 这个电视剧的男主男一号叫靳燃。
The male protagonist main male character of this drama is named Jin Ran. | 这个电视剧的男一号叫靳燃。
The main male character of this drama is named Jin Ran. | Disfluency |
| Prompt 1 Demonstration 4 | 腿可以够到很多你胳膊够不到的地方。
The leg can reach a lot of places your arm can't. | 腿可以碰到很多你胳膊碰不到的地方。
The leg can touch a lot of places your arm can't. | Colloquial Word |
| Prompt 1 Demonstration 5 | 他消失了二十八年他终于研究出了那个就是氢弹原子弹这些。
He disappeared for 28 years and he finally developed the like the, the hydrogen bombs and atomic bombs. | 他消失了二十八年他终于研究出了氢弹原子弹这些武器。
He disappeared for 28 years and he finally developed weapons like hydrogen bombs and atomic bombs. | Disfluency and Grammatical Error |
| Prompt 2 Demonstration 1 | 人家会做什么一大桌桌子的菜。
They will cook, uh, a whole ta-, table of food. | 人家会做一大桌子的菜。
They will cook a whole table of food. | Disfluency |
| Prompt 2 Demonstration 2 | 这就是职业带来的好一些好处。
These are some of the be-, benefits that come with a career. | 这就是职业带来的一些好处。
These are some of the benefits that come with a career. | Disfluency |
| Prompt 2 Demonstration 3 | 你我们理性分析的话会发现他很聪明。
If you, we analyze him rationally, we will find that he is very clever. | 我们理性分析的话会发现他很聪明。
If we analyze rationally, we will find that he is very clever. | Disfluency |
| Prompt 2 Demonstration 4 | 你你就算吃饭了还是会饿。
You you will still be hungry even after eating. | 你就算吃饭了还是会饿。
You will still be hungry even after eating. | Disfluency |
| Prompt 2 Demonstration 5 | 她她师傅是凭着真实力取得了冠军。
Her, her master won the championship with real strength. | 她师傅是凭着真实力取得了冠军。
Her master won the championship with real strength. | Disfluency |
| Prompt 3 Demonstration 1 | 除了除了选择工作以外，还有很多人继继续学习深造。
In addition, in addition to work, many people continue their studies. | 除了选择工作以外，还有很多人继续学习深造。
In addition to work, many people continue their studies. | Disfluency |
| Prompt 3 Demonstration 2 | 你喜不喜欢就是旅游呀啥的？
Do you like, uh, traveling or not? | 你喜不喜欢旅游？
Do you like traveling? | Disfluency |
| Prompt 3 Demonstration 3 | 张三丰这个人是有史料可卡找的。
Zhang Sanfeng has historical materials to trap find. | 张三丰这个人是有史料可找的。
Zhang Sanfeng has historical materials to find. | Grammatical Error |
| Prompt 3 Demonstration 4 | 让我印象最深的一次一次兼职就是快递快递分拣。
One, one of the most impressive part-time jobs I've ever had was courier, courier sorting. | 让我印象最深的一次兼职就是快递分拣。
One of the most impressive part-time jobs I've ever had was courier sorting. | Disfluency |
| Prompt 3 Demonstration 5 | 我习惯了习惯了早睡早起。
I'm getting used to, used to going to bed early and getting up early. | 我习惯了早睡早起。
I'm used to going to bed early and getting up early. | Disfluency |

Table 12: Demonstrations used in the three prompts.

| Prompt | Model | BLEU-1 | BLEU-2 | BLEU-3 | BLEU-4 |
|--------|-------|--------|--------|--------|--------|
| 0 | BELLE-7B-2M | 0.4885 | 0.3519 | 0.2790 | 0.2167 |
|   | ChatGLM-6B | 0.5924 | 0.4872 | 0.4261 | 0.3785 |
|   | GPT3.5-turbo | 0.6882 | 0.5862 | 0.5087 | 0.4393 |
| 1 | BELLE-7B-2M | 0.4593 | 0.3465 | 0.2688 | 0.2042 |
|   | ChatGLM-6B | 0.6600 | 0.5445 | 0.4596 | 0.3847 |
|   | GPT3.5-turbo | 0.6260 | 0.5215 | 0.4442 | 0.3748 |
| 2 | BELLE-7B-2M | 0.4558 | 0.3532 | 0.2790 | 0.2166 |
|   | ChatGLM-6B | 0.6932 | 0.5977 | 0.5225 | 0.4530 |
|   | GPT3.5-turbo | 0.6918 | 0.5971 | 0.5249 | 0.4586 |
| 3 | BELLE-7B-2M | 0.4559 | 0.3598 | 0.2910 | 0.2312 |
|   | ChatGLM-6B | 0.6842 | 0.5822 | 0.5043 | 0.4324 |
|   | GPT3.5-turbo | 0.7331 | 0.6131 | 0.5333 | 0.4644 |

Table 13: The BLEU-1, BLEU-2, BLEU-3, and BLEU-4 scores using BELLE-7B-2M, ChatGLM-6B, and GPT3.5-turbo, each with the three prompts. To facilitate comparison, we include the results from Section 5.4 in this table, and we refer to the prompt used in Section 5.4 as "Prompt 0". The results of "Prompt 0" are considered the baseline, and any improvements over Prompt 0 are highlighted in orange, while decreases are marked in blue.