# OpenReview forum: "CS2W: A Chinese Spoken-to-Written Style Conversion Dataset with Multiple Conversion Types"
_EMNLP/2023/Conference — EMNLP 2023 Main_

### Official Review · Reviewer_i9a8 · 2023-08-03

**Soundness:** 5

**Excitement:**

5: Transformative: This paper is likely to change its subfield or computational linguistics broadly. It should be considered for a best paper award. This paper changes the current understanding of some phenomenon, shows a widely held practice to be erroneous in someway, enables a promising direction of research for a (broad or narrow) topic, or creates an exciting new technique.

**Missing References:**

line 329 - PKU seg (available at https://arxiv.org/abs/1906.11455)

line 425 - BLOOM (cite: https://arxiv.org/abs/2211.05100)

line 435 - ChatGLM (cite either: https://arxiv.org/abs/2103.10360 for the 6B model or https://arxiv.org/abs/2210.02414 for the 130B model)

**Paper Topic And Main Contributions:**

This paper describes the problem of taking transcribed Chinese text and performs a style transformation to remove disfluencies and grammatical errors, presenting an output that is more consistent with written style. To solve this problem they have prepared an open-source (non-commercial) corpus of annotated data based on an existing dataset, they have fine-tuned several existing models to establish baselines for this dataset, and have demonstrated the utility of the dataset in addressing a downstream machine translation task.

**Questions For The Authors:**

Question A: Regarding ASR transcription errors, you have chosen a single source of ASR transcriptions. What confidence should we have that this ASR transcriber produces errors in a similar manner to other ASR systems?

Question B: For the zero to five demonstrations given to the LLMs, were they selected at random? Did they contain the same type of error as you expected the model to find? When you have more demonstrations, did you ensure diversity in the examples given to the LLM?

Question C: Did the annotators give informed consent for the use of their contributions in training AI/NLP systems?

**Reasons To Accept:**

This paper sets a standard for addressing multiple types of errors in spoken transcription through one model, with an adequate-sized corpus for future academic work.The paper appears to be free of major methodological errors, demonstrates the utility of multiple large language models against the task, and follows up by demonstrating a substantial improvement on a downstream task.

**Reasons To Reject:**

There are some errors in visualization, some typos, and some missing citations, but none of these rise to the level of being reasons to reject this paper.



**Reproducibility:**

5: Could easily reproduce the results.

**Reviewer Confidence:**

4: Quite sure. I tried to check the important points carefully. It's unlikely, though conceivable, that I missed something that should affect my ratings.

**Typos Grammar Style And Presentation Improvements:**

line 030 - "canonicalized" can just be "canonical"

line 039 - Please indicate "Mandarin Chinese" not just "Chinese" at the first mention. It is okay to use "Chinese" subsequently, where it will be understood to mean Mandarin, but it may be clearer to use Mandarin throughout.

line 383 - "Figure" should say "Figure 3"

line 383-385 - would read more clearly with "said incorrectly" in place of "incorrectly"

section beginning line 412 - please indicate the parameter count of the various models used, so their performance can be measured in comparison with their parameter efficiency

Figure 2 - the graph on the left should start at zero, not one. The graph on the right has three bars labeled "2" that are of different lengths

Figure 4 - the sentence at the bottom should say "is not" not "is not only" based on the transcript above. Also ... in this example, the typical correct English would be either "... is not only for pandas, ..." or "... is not for pandas alone, ..." The example might confuse English readers (even if it is correct for Chinese)

line 465 - "5 - shot" in this context could either mean five examples were provided in the prompt or that five generations were used (assuming a temperature that is not fully deterministic). It may help to clarify that five examples were provided, and to specify the model temperature for generation.

line 475 - you have been using the label "colloquial words" for this category, and you change to "spoken words" here. It should be consistent, and "colloquial words" is the better choice.

Table 7 - the description says BLEU 1, 2, and 3 the labels and text say 1, 2, and 4. Also, the header labels are inconsistent. This table is also presented out of order for the numbering

Table 6 - GPT should be CPT in the top left. Also, use "Colloquial Words" not "Spoken words" here

---

> ### Author Rebuttal · Authors · 2023-08-29
>
> Thank you very much for your insightful comments and suggestions. We are very encouraged by your positive comments on our work.
>
> **Q1**: Regarding ASR transcription errors, you have chosen a single source of ASR transcriptions, What confidence should we have that this ASR transcriber produces errors in a similar manner to other ASR systems?
>
> **A1**: Thank you for bringing this issue to our attention. We have double-checked the MagicData-RAMC dataset and the description of data construction, and found no information about the ASR system used for transcription of this dataset.
>
> **Q2**: For the zero to five demonstrations given to the LLMs, were they selected at random? Did they contain the same type of error as you expeted the model to find? When you have more demonstrations, did you ensure diversity in the examples given to the LLM?
>
> **A2**: Thank you for your questions.
>
> (1) Yes, the zero to five demonstrations given to the LLMs were chosen at random. There are four types of conversion problems in CS2W. As we expected the LLMs to be able to correct all types of conversion problems and the conversion problems presented in the demonstration may enhance the understanding of LLMs towards the corresponding conversion problems, we enforced diversity in the demonstrations given to LLMs during the demonstration selection process. Considering that there are four types of conversion problems in CS2W, we continued to add demonstrations of different types until there was no room to add more demonstrations. During the demonstration selection process, the demonstrations of each conversion problem were randomly selected from the validation set, which were different from the input sentence in the test set and contained only one conversion problem. In addition, CS2W is dominated by the disfluency (87.4%) conversion problem, so we selected two demonstrations which were R-type and Filter word respectively under the 5-shot setting. In sections 5.4 and 5.7, the prompt used in 5-shot experiments is “下面有一个语法纠错任务，请把错误的文本修改为正确的文本：\\n\\n错误文本：我不不太喜欢听那种歌曲；\\n正确文本：我不太喜欢听那种歌曲。\\n\\n错误文本：你呃喝酒喝的比较多了；\\n正确文本：你喝酒喝的比较多了。\\n\\n错误文本：我爸会糖醋丸子；\\n正确文本：我爸会炸糖醋丸子。\\n\\n错误文本：我想给游戏氪金；\\n正确文本：我想给游戏充钱。\\n\\n错误文本：这件事情我无可奉报；\\n正确文本：这件事情我无可奉告。\\n\\n错误文本：{源句}\\n正确文本：\\n” (Here's a grammatical error correction task, please correct the wrong text into the right text: \\n\\n wrong text: I don’t don’t like this type of music;\\n correct text: I don’t like this type of music. \\n\\n wrong text: You, uh, have been drinking a lot;\\n correct text: You have been drinking a lot.\\n\\n wrong text: My dad can sweet and sour dumplings;\\n correct text: My dad can make sweet and sour dumplings.\\n\\n wrong text:  I'd like to load up the game;\\n correct text: I'd like to top up the game.\\n\\n wrong text: I have nothing to report in this matter;\\n correct text: I have nothing to say in this matter. \\n\\n wrong text: {source sentence}\\n\\n correct text: ).
>
> (2) Yes, when there were five demonstrations, they contained the same type of error as we expected the model to find. As we added demonstrations with disfluency conversion problems first and there were two demonstrations with disfluency conversion problems, if the number of demonstrations was less than five, we also attempted to enforce diversity in the demonstrations by continually adding demonstrations with different conversion problems until there was no room to add more demonstrations.
>
> (3) Yes, we indeed ensured diversity in the examples given to LLMs during the demonstration selection process. Please refer to the detailed demonstration selection process described above for more information. However, although we had an intuition that having more types of conversions in demonstrations would enable LLMs to resolve more types of conversions and improve results, we found that this might not hold true under all conditions. We will provide a detailed illustration of the experiment below:
>
> For convenience, the results in section 5.4 is shown in the table below.
>
> | Model\BLEU | BLEU-1 | BLEU-2 | BLEU-3 | BLEU-4 |
> | :------: | :---: | :---: | :---: | :---: |
> |  BELLE-7B-2M   |   0.4885    |   0.3519    |    0.279   |    0.2167   |
> | ChatGLM-6B  |    0.5924   |    0.4872   |    0.4261   |   0.3785    |
> |  GPT3.5-turbo  |   0.6882    |    0.5862   |    0.5087   |    0.4393   |
>
> We randomly selected three groups of five demonstrations from the validation set:
>
> The first group:
>
> demonstration1:(colloquial word)
>
> wrong text: 我为什么要套路女孩子; correct text: 我为什么要哄骗女孩子。
>
> wrong text: Why I'm trying to trap a girl. correct text: Why I'm trying to cajole a girl.
>
> demonstration 2: (disfluency)
>
> wrong text: 他他让我假装睡觉; correct text: 他让我假装睡觉。
>
> wrong text: He he asked me to pretend to sleep; correct text: He asked me to pretend to sleep.
>
> demonstration 3: (disfluency)
>
> wrong text: 这个电视剧的男主男一号叫靳燃;correct text: 这个电视剧的男一号叫靳燃。
>
> wrong text: The male protagonist main male character of this drama is named Jin Yan; correct text: The main male character of this drama is named Jin Yan.
>
> demonstration 4: (colloquial word)
>
> wrong text: 腿可以够到很多你胳膊够不到的地方;correct text: 腿可以碰到很多你胳膊碰不到的地方。
>
> wrong text: the leg can reach a lot of places your arm can't; correct text: the leg can touch a lot of places your arm can't.
>
> demonstration 5: (disfluency, grammatical error)
>
> wrong text: 他消失了二十八年他就终于研究出了那个就是氢弹原子弹这些; correct text: 他消失了二十八年他就终于研究出了氢弹原子弹这些武器。
>
> wrong text: He disappeared for 28 years and he finally developed the, the, hydrogen bomb and the atomic bomb and so on;
> correct text: He disappeared for 28 years and he finally developed the hydrogen bomb and the atomic bomb.
>
> The results of the demonstrations for the first group are presented in the table below:
>
> | Model\BLEU | BLEU-1 | BLEU-2 | BLEU-3 | BLEU-4 |
> | :------: | :---: | :---: | :---: | :---: |
> |  BELLE-7B-2M   |   0.4593  $\downarrow$  |   0.3465  $\downarrow$ |    0.2688 $\downarrow$  |    0.2042  $\downarrow$ |
> | ChatGLM-6B  |    0.66 $\uparrow$   |    0.5445  $\uparrow$  |    0.4596 $\uparrow$  |   0.3847 $\uparrow$   |
> |  GPT3.5-turbo  |   0.626  $\downarrow$  |    0.5215  $\downarrow$ |    0.4442 $\downarrow$  |    0.3748 $\downarrow$  |
>
> The results indicate that the BLEU score of ChatGLM-6B increased slightly, while the BLEU scores of the other two LLMs decreased.
>
> The second group with five demonstrations with disfluency and only one conversion type:
>
> demonstration1:(disfluency)
>
> wrong text: 人家会做什么一大桌桌子的菜; correct text: 人家会做一大桌子的菜。
>
> wrong text: They cook a whole table and a table of food; correct text: They cook a whole table of food.
>
> demonstration 2: (disfluency)
>
> wrong text: 这就是职业带来的好一些好处; correct text: 这就是职业带来的一些好处。
>
> wrong text: These are some of the be- benefits that come with a career. correct text: These are some of the benefits that come with a career.
>
> demonstration 3: (disfluency)
>
> wrong text: 你我们理性分析的话会发现他很聪明; correct text: 我们理性分析的话会发现他很聪明。
>
> wrong text: If you, we analyze him rationally, we will find that he is very clever. correct text: If we analyze rationally, we will find that he is very clever.
>
> demonstration 4: (disfluency)
>
> wrong text: 你你就算吃饭了还是会饿; correct text: 你就算吃饭了还是会饿。
>
> wrong text: You you will still be hungry even if you eat; correct text: You will still be hungry even after eating.
>
> demonstration 5: (disfluency)
>
> wrong text: 她她师傅是凭着真实力取得了冠军; correct text: 她师傅是凭着真实力取得了冠军。
>
> wrong text: She, her master won the championship by virtue of true strength; correct text: Her master won the championship with real strength.
>
> The results of the demonstrations for the second group are presented in the table below:
>
> | Model\BLEU | BLEU-1 | BLEU-2 | BLEU-3 | BLEU-4 |
> | :------: | :---: | :---: | :---: | :---: |
> |  BELLE-7B-2M   |   0.4558  $\downarrow$  |   0.3532 $\uparrow$  |    0.279 $\uparrow$  |    0.2166 $\downarrow$  |
> | ChatGLM-6B  |    0.6932 $\uparrow$ |   0.5977 $\uparrow$  |    0.5225 $\uparrow$  |    0.453 $\uparrow$   |
> |  GPT3.5-turbo  |   0.6918   $\uparrow$ |    0.5971  $\uparrow$ |    0.5249  $\uparrow$ |    0.4586  $\uparrow$ |
>
> The results indicate that the BLEU score of BELLE-7B-2M is consistent with the results presented in section 5.4, while the BLEU scores of the other two LLMs increased significantly.
>
> The third group with four demonstrations with disfluency and two conversion type:
>
> demonstration1: (disfluency)
>
> wrong text: 除工除了选择工作以外，还有很多人继续学习深造；correct text: 除了选择工作以外，还有很多人继续学习深造。
>
> wrong text: In addition, in addition to choosing a job, many people continue their studies; correct text: In addition to choosing jobs, many people continue their studies.
>
>
> demonstration 2: (disfluency)
>
> wrong text: 你喜不喜欢就是旅游呀啥的；correct text: 你喜不喜欢旅游呀。
>
> wrong text: 你喜不喜欢就是旅游呀啥的; correct text: 你喜不喜欢旅游呀。
> wrong text: Do you like , uh, traveling or not? correct text: Do you like traveling?
>
> demonstration 3: (grammatical error)
>
> wrong text: 张三丰这个人是有史料可卡找的; correct text: 张三丰这个人是有史料可找的。
>
> wrong text: Zhang Sanfeng has historical materials to trap and find; correct text: Zhang Sanfeng has historical materials to find;
>
> demonstration 4: (disfluency)
>
> wrong text: 让我印象最深的一次一次兼职就是快递快递分拣；correct text: 让我印象最深的一次兼职就是快递分拣。
>
> wrong text: One one of the most impressive part-time job I've ever had was courier, courier sorting; correct text: One of the most impressive part-time job I've ever had was courier sorting.
>
> demonstration 5: (disfluency)
>
> wrong text: 我习惯了习惯了早睡早起；correct text: 我习惯了早睡早起。
>
> wrong text: I'm used to getting used to going to bed early and getting up early; correct text: I'm used to going to bed early and getting up early.
>
>
> The results of the demonstrations for the third group are presented in the table below:
>
> | Model\BLEU | BLEU-1 | BLEU-2 | BLEU-3 | BLEU-4 |
> | :------: | :---: | :---: | :---: | :---: |
> |  BELLE-7B-2M   |   0.4559 $\downarrow$   |   0.3598  $\uparrow$ |    0.291 $\uparrow$  |    0.2312 $\uparrow$  |
> | ChatGLM-6B  |    0.6842 $\uparrow$ |   0.5822 $\uparrow$  |    0.5043 $\uparrow$  |    0.4324  $\uparrow$  |
> |  GPT3.5-turbo  |   0.7331  $\uparrow$  |    0.6131 $\uparrow$  |    0.5333  $\uparrow$ |    0.4644 $\uparrow$  |
>
> The results indicate that the BLEU scores of all LLMs increased.
>
> In summary, the results of the experiment indicate that although we surmised that greater diversity in examples would enable LLMs to resolve more types of conversions and improve results, this was not always the case. In fact, the first group, which had the greatest diversity in examples, performed the worst. The second and third groups, which had less diversity but more demonstrations with disfluencies, performed better. We surmise that this is because disfluency dominates CS2W, and when the number of demonstrations with disfluencies exceeds a certain threshold, the results improve. If the four conversion types in the dataset were balanced, demonstrations with greater diversity may have performed better.
>
> We will provide more details on the prompts and outputs of LLMs in the next version.
>
> **Q3**: Did the annotators give informed consent for the use of their contributions in training AI/NLP systems?
>
> **A3**: Yes, the annotators gave informed consent for the use of their contributions in training AI/NLP systems.
>
> **Typos Grammar Style And Presentation Improvements**:
>
> **I1**: Section beginning line 412 - please indicate the parameter count of the various models used, so their performance can be measured in comparison with their parameter efficiency.
>
> **A1**: Thank you for your feedback. The parameter count of the various models used is shown in the table below.
>
> |      Model      | BART-base | BART-large | CPT-base | CPT-large | BLOOM | BELLE | ChatGLM |
> | :-------------: | :-------: | :--------: | :------: | :-------: | :---: | :---: | :-----: |
> | **Parameter Count** |   139M    |    406M    |   121M   |   393M    |  7B   |  7B   |   6B    |
>
> **I2**: Line 465 - "5 - shot" in this context could either mean five examples were provided in the prompt or that five generations were used (assuming a temperature that is not fully deterministic). It may help to clarify that five examples were provided, and to specify the model temperature for generation.
>
> **A2**: Thank you for this comment. The “5-shot” in this context means five examples were provided in the prompt.
>
> Regarding the prompts used for zero- and few-shot setting, we tried two different prompts on BLOOM-7B under the zero-shot and 5-shot setting.
>
> Prompt1: “下面有一个口语到书面语风格转换任务，请把口语修改为书面语：\\n\\n口语：\\n {源句} \\n书面语：\\n”
>
> (Here's a spoken- to written-language style conversion task, please rewrite the spoken language into the written language: \\n\\n Spoken: {source sentence} \\n Written: \\n).
>
> Prompt2: "下面有一个语法纠错任务，请把错误的文本修改为正确的文本：\\n\\n错误文本：\\n{源句} \\n 正确文本：\\n"
>
> (Here's a grammatical error correction task, please correct the wrong text into the right text: \\n\\n wrong text: {source sentence} \\n correct text: \\n)
>
> Prompt2 outperforms Prompt1 on all metrics. We speculate that the model is not capable of understanding the definition of the spoken-to-written language conversion task. However, grammatical error correction is a widely-used task and the four conversion problems, except for colloquial words, can be regarded as either simple or complex grammatical errors, thus making it work well. Hence we adopt prompt2 for the rest of our experiments.
>
> Under the zero-shot setting, the prompt is “下面有一个语法纠错任务，请把错误的文本修改为正确的文本：\\n\\n错误文本：\\n {源句} \\n正确文本：\\n” (Here's a grammatical error correction task, please correct the wrong text into the right text: \\n\\n wrong text: {source sentence} \\n\\n correct text:).
>
> The five examples please refer to the response for Q2.
>
> The model temperature for generation is shown in the table below.
>
> |    Model    | BLOOM-7B | BELLE-7B-0.2M | BELLE-7B-2M | ChatGLM-6B | GPT3.5-turbo |
> | :---------: | :------: | :-----------: | :---------: | :--------: | :----------: |
> | **Temperature** |   1.0    |     0.35      |     1.0     |    0.95    |     0.7      |
>
> **I3**: Table 7 - the description says BLEU 1, 2, and 3 the labels and text say 1, 2, and 4. Also, the header labels are inconsistent. This table is also presented out of order for the numbering.
>
> **A3**: Many thanks for this. The correct caption is “BLEU-1, BLEU-2, BLEU-4”. We will change it in the next version.
>
> Other typos, grammatical errors and presentation issues will be fixed all in the next version. Many thanks again.

---

### Official Review · Reviewer_3uWx · 2023-08-04

**Soundness:** 4

**Excitement:**

4: Strong: This paper deepens the understanding of some phenomenon or lowers the barriers to an existing research direction.

**Missing References:**

For Baoli (2011), Shriberg (1996) and Zayats et al. (2016), may you add where the papers were published (e.g. arxiv)?

**Paper Topic And Main Contributions:**

The paper addresses the conversion of spoken-style language to written-style language as a postprocessing step for automatic speech recognition (ASR). The main contributions are the creation and publication of a manually annotated dataset for Chinese and benchmark evaluations with large language models (LMs) on this dataset.

The dataset is a manually collected subset of the Real Spontaneous Dialogue Speech dataset MagicData-RAMC by Yang et al. (2022) and consists of 7237 samples containing relevant spoken-to-written-style issues. Each sample is annoted with the types and spans of issues, together with a manually normalized written-style version. Four different types of issues are annotated: disfluencies (repetitions, restarts, and repairs), ASR transcription errors, grammatical errors, and spoken-style vocabulary. According to the authors, it is the first dataset with annotation of multiple error types for Chinese.

In addition, the paper analyzes six different LMs to benchmark their performance on the Chinese spoken-to-written conversion task: BART, CPT, BLOOM, BELLE, ChatGLM and GPT3.5. Only the first two are actually trained on the curated dataset. For the others, both zero-shot and few-shot (5-shot) settings are tested. The result shows that CPT-large, trained on the curated dataset, has the best performance across various metrics.

**Questions For The Authors:**

1) In section 4.4, you describe a new sub-type of disfluency. Can you quantify how common this type of disfluency is?

2) Do I understand correctly that you do not use the originally created training, development and test sets (described on lines 323f.) in your experiments (because you use a different split described on lines 400f.)? If so, what is the benefit of the original splits?

3) In Figure 4, the English translation of the example is confusing: In the first line, you translate 不单 as "is not only" and 不单独 as "is not". However, in the following lines, both versions are translated as "is not only". Can you explain this inconsistency? Also, you do not translate 吃. How about, e.g., "Bamboo is not only a food for pandas, but it can also be made into a lot of furniture."?

4) Could you please show/include the prompts you used for zero-shot and few-shot inference?

5) Are there similar datasets in other languages than EN, CH, JA and VI, or did you focus on purpose on this selection of languages?

6) On line 198, you say that CS2W offers two annotation paradigms. Could you clarify what you mean here?

7) Figure 3 b): Unfortunately, I don't understand this figure. Is it the number of spans per samples? (If so, I think you can remove the colors and the labels, as this seems to be redundant.)

**Reasons To Accept:**

The authors create and publish a dataset for a task for which few public datasets exist to date. Unlike other similar datasets, they annotate multiple style conversions in parallel (disfluencies, ASR transcription errors, grammatical errors, and spoken-style vocabulary), which makes it suitable for real-world scenarios. Ideally, their corpus may serve as a blueprint for similar datasets for languages other than Chinese.

Their experiments show that LMs finetuned on this dataset can outperform zero-shot and few-shot models on several metrics, demonstrating the usefulness of such a manually curated dataset even in the era of pretrained LMs.

**Reasons To Reject:**

The paper does not describe the creation of the corpus in full detail. For example, it does not explain how the samples were selected from the larger MagicData RAMC of Yang et al. (2022), and it does not describe the annotation guidelines in detail. While it is mentioned that there were inter-annotator disagreements, no examples of these are given, nor is it explained how the disagreements were resolved.

In addition, most experiments use zero-shot or few-shot settings, disregarding the dataset created. Although the performance of models finetuned on the dataset is indeed better, this point is not sufficiently emphasized or discussed. Again, some details are missing, e.g., which prompts were used for zero-shot and few-shot experiments.

**Reproducibility:**

4: Could mostly reproduce the results, but there may be some variation because of sample variance or minor variations in their interpretation of the protocol or method.

**Reviewer Confidence:**

4: Quite sure. I tried to check the important points carefully. It's unlikely, though conceivable, that I missed something that should affect my ratings.

**Typos Grammar Style And Presentation Improvements:**

For all Chinese texts, I'd recommend to also add Pinyin (including diacritics) for accessibility.

Table 2:
- NLPCC18-task: "Rewriting"> "Direct Rewriting" (for consistenc with other lines, unless there is a difference in which case please clarify)
- PhoDisfluency: "JA" > "VI" (Vietnamese)
- "HSK" > "HSK Exam" (like "TOEFL Exam")

Table 3:
- "The milk is a little bit very hot." Isn't this an example of an R-type disfluency instead of a redundant word?
- "Incorrect Word / Order": The word "Order" is in the next cell, which is confusing

Table 4:
- "w/ 2 error" > "w/ 2 errors" (and accordingly for 3 and 4 errors)

Table 5:
- "Bloom-7b" > "BLOOM-7B"
- The table would be clearer if you would arrange it into three parts: 1) fine-tuning, 2) zero-shot, 3) 5-shot

Table 6: "GPT-large" > "CPT-large"

Table 7: "BLEU.2" > "BLEU-2"

Figure 1: In the caption, you have the abbreviation FP=Filled Pause. However, this does not appear in the example. Or should FW be FP?

Figure 3:
- "conversion" > "Conversion"
- maybe change order of categories, e.g. from small to large (in this case, switch the first two, and switch GE and CW further down)

155: Can you give an example of collocation errors?

156: remove "which"

178: What do you mean by "doesn't require annotators"?

189: remove "Pinyin of"

247: "&" > "and"

267: "by Sakaguchi et al. (2016)" (i.e., change citation type)

277f: This sentence is incorrectly constructed and confusing, please check it

311f: This sentence is unclear to me.

347: remove "much"

355: "exhibit" > "our dataset exhibits"

361: "prolems" > "problems"

366f: "with disfluency/grammatical errors …": This sentence is unclear to me, isn't it a repetition of the main sentence ("Among the mixed types …")?

383: "Figure" > "Figure 4"

424: "We used cpt-base" > "We used cpt-base and cpt-large"

426f: add reference for the ROOTS corpus

445: "We test fine-tuning with […] and zero-shot and 5-shot with BLOOM, BELLE, ChatGLM, and GPT3.5-turbo.

452: Please explain what the P/R/F scores are.

---

> ### Author Rebuttal · Authors · 2023-08-29
>
> Thank you very much for your insightful comments and suggestions.
>
> **Reason to reject:**
>
> **Q1:** The paper does not describe the creation of the corpus in full detail. For example, it does not explain how the samples were selected from MagicData RAMC, and it does not describe the annotation guidelines in detail, and it does not explain how the disagreements were resolved.
>
> **A1:** We will describe data extraction, detailed labeling guidelines, and how to resolve disagreements below.
> 1. Data Extraction
>
>     MagicData-RAMC is an ASR transcription dataset consisting of 351 sets of spontaneous conversations in Chinese Mandarin. Each set features natural conversations between two speakers on a single topic, and it includes audio files and transcribed texts that retain disfluencies, grammatical errors, and ASR transcription errors. We manually select sentences from the ASR transcriptions, which are self-contained in meaning but with conversion problems for annotation.
>
>     In transcribed text, we do not extract the following three types of sentences:
>
>     (1) Incomplete sentences. In spontaneous conversations, a speaker often break off abruptly or is interrupted by another speaker, resulting in many incomplete sentences in transcriptions.
>
>      e.g. 我和他刚刚准备出去玩，就(He and I were just getting ready to hang out but)
>
>     This indicates that the speaker stops speaking at the word "but" or is interrupted by someone else. Incomplete sentences cannot be understood by annotators and hence cannot be accurately annotated.
>
>     (2) Sentences that require context. Spontaneous conversations have continuity and many sentences need to be understood in context.
>
>     e.g.
>
>     Speaker1：你这次数学考试考了多少分？(Speaker1：What grade did you get on this math test? )
>
>     Speaker2：一百多一点，比不上你。(Speaker2：A little over a hundred, no more than you. )
>
>     Speaker2's speech is a response to Speaker1's question, which is not understandable without Speaker1’s speech. The sentences that require context also can hardly be annotated.
>
>     (3) Sentences that are too short. In spoken language, phrases such as "no problem", "yes" and "okay" are often used, which are too general. Therefore, we do not select sentences with the number of tokens being less than 5.
>
>     In conclusion, we select complete sentences with conversion problems whose length are appropriate.
>
> 2. Detailed Annotation Guidelines
>
>     (1) Task Statement
>
>     Automatic speech recognition (ASR) plays a vital role in a wide range of NLP application scenarios. Spoken language is transcribed into spoken style texts which serve as fundamental inputs to plenty of downstream tasks. However, they are often inherently containing disfluencies, grammatical errors, and colloquial words. This dataset consists of transcribed texts with conversion problems for annotation. For each sentence, the annotator needs to annotate the type and the range of conversion problems, and write the corresponding written language.
>
>     (2) Conversion type definitions and examples
>
>     We further categorize the conversion problem with a two-level classification system as: Disfluency (R-type, Filter Word), ASR Transcription Errors, Grammatical Errors (Missing Words, Redundant Words, Incorrect Word Order), Colloquial Words.
>
>     Each conversion problem is described in detail next.
>
>     ① Disfluency
>
>     The elements that make a sentence not fluent are referred to as "disfluency", which can be categorized into R-type and Filter Word based on their structures [4].
>
>     **Filter Word:**
>
>     Filter word have no meaning and are often used to indicate pauses and hesitations of the speaker, which includes "uh", "ah" and so on. Many times, words like "yeah", "okay" are also marked as filler words.
>
>     e.g. Well, this is, you know, a good plan.
>
>     In this sentence, the phrases "well" and "you know" have no meaning, and they should be annotated as "Filter Word".
>
>     **R-type:**
>
>     The standard structure of R-type disfluency indicates the reparandum (words that the speaker intends to discard), an optional interregnum (words that have no meaning), and the associated repair. Interregnums are often fixed phrases, e.g. "uh", "you know".
>
>     Reparandum contains those words which are originally not intended to be in the utterance. Thus this section consists of one or more words that will be repeated or corrected. Interregnum is a non-lexicalized part like filter word, including "uh", "I mean" and so on. The last part is repair. Words from the reparandum are finally corrected or repeated.
>
>     e.g. 让我们，我的意思是，让我来解决这个问题。(Let us, I mean, let me work on the problem.)
>
>     In this sentence, “Let us” is the reparandum, and “let me” is the repair, which corrects the reparandum. "I mean" have no meaning so it is the interregnum
>
>     ② ASR Transcription Errors
>
>     ASR Transcription Errors are the occasional homophone mistakes in ASR transcriptions. CS2W is built upon the existing MagicData-RAMC dataset, which consists of 351 sets of spontaneous conversations speech in Chinese Mandarin and their ASR transcriptions. Thus there are some occasional homephone mistakes in some sentences.
>
>     e.g. 这个艺术家很有菜花。(The artist is very cauliflower.)
>
>      (footprint: In Chinese, "cauliflower" and "talent" have the same pronunciation.)
>
>     According to the meaning of the sentence, the correct sentence should be "The artist is very talented". So "cauliflower" needs to be annotated as “ASR Transcription Error”.
>
>     ③ Grammatical Errors
>
>     The transcription text in spoken language often includes grammatical errors because speakers in conversations often lack careful thinking. And common grammatical errors in spoken language include Missing Words, Redundant Words, and Incorrect Word Order.
>
>     **Missing Words:**
>
>     Missing Words include missing subject, missing predicate, missing object, missing function word, and missing modifier.
>
>     e.g. 那时我们有机会扳平比分，但是我们没有机会。(We had a chance to equalize, but we didn't it.)
>
>     This sentence is missing a verb. The correct sentence is "We had a chance to equalize, but we didn't take it".
>
>     **Redundant Words:**
>
>     Redundant Words include redundant subject, redundant predicate, redundant object, redundant function word, and redundant modifier.
>
>     e.g. 它们的皮毛很有光泽,可以用肉眼很难看出来。(Their fur is shiny and can be hardly seen with the naked eyes.)
>
>     The modifier in this sentence is redundant. The fur is shiny so it should be visible to the naked eyes. The word "hardly" should be deleted.
>
>     **Incorrect Word Order:**
>
>     Incorrect Word Order is also common in spoken transcribed texts because of the frequent inversions in spoken language.
>
>     e.g. 昨天我看了新买的一部电影在电视上。(Yesterday I watched a newly purchased movie on TV.)
>
>     In Chinese, the correct sentence is "Yesterday on TV I watched a newly purchased movie".
>
>     ④ Colloquial Words
>
>     Spoken language often contains informal expressions, such as some popular Internet phrases, which is called “Colloquial Words”.
>
>     e.g. 这明明是你的功劳，却被同事抢走了，你真是一个大怨种。(This is obviously your credit, but your coworkers took it away. You're such an unlucky guy.)
>
>     (In Chinese, "大怨种" is a popular Internet phrase which is used to describe people who are aggrieved but have no way to complain.)
>
>     We need to replace all informal expressions with formal ones. The written language is "This is obviously your credit, but your coworkers took it away. You're such an unlucky guy".
>
>     ⑤ Mixed Type
>
>     In real spontaneous conversations, a single sentence often contains multiple conversions.
>
>     e.g. 在国内成立野牛，这个，水牛研究中心，有利于帮助适应人工环境。(The establishment of the bison, I mean, buffalo research center in the country will help to adapt in an artificial environment.)
>
>     This sentence contains both disfluency and grammatical errors. The written language should be “The establishment of the buffalo research center in the country will help them to adapt in an artificial environment.”.
>
>     (3) Annotation rules
>
>     We built an annotation platform to accelerate our annotation progress. When using it, the annotator needs to select the conversion type and annotation range of the current sentence. Then, the annotator provides the written language corresponding to this spoken language. Next we will present the annotation rules for different conversions. Please note that on the annotation platform, each sentence is word-segmented into individual words, each of which can be selected to make it easier for the annotator to annotate the conversion range.
>
>     ① Disfluency
>
>     The annotator selects the disfluency button on the annotation platform. For Filter Word, the annotator should annotate its range. For R-type, the reparandum and the interregnum should be annotated but the repair does not need to be annotated. Because the sentence can be corrected with the reparandum and the interregnum being deleted.
>
>     ② ASR Transcription Errors
>
>     The annotator selects the ASR transcription errors button on the annotation platform and annotates the range of the ASR Transcription Errors.
>
>     ③ Grammatical Errors
>
>     The annotator selects the grammatical errors button on the annotation platform. For Missing Words, the annotator needs to annotate the two words before and after the missing part. For Redundant Words, the annotator needs to annotate the redundant part. For Incorrect Word Order, the whole sentence has to be annotated.
>
>     ④ Colloquial Words
>
>     The annotator selects the colloquial words button on the annotation platform and annotates the range of the Colloquial Words.
>
>     ⑤ Mixed Type
>
>     First, the annotator selects the button for the first conversion type, and then annotates the range of this conversion. Then the annotator selects the buttons of the other conversion types in turn and annotate the range of the conversion.
>
>     (4) Disagreement Resolution
>
>     To guarantee annotation consistency and quality, we introduce a two-round annotation process.
>
>     In the first round of annotation, we recruit eight Chinese native speakers as part-time annotators to start annotation after pre-annotation training.
>
>     In the second round, we perform manual evaluation and re-annotation with other two annotators, and one of the authors of this paper serves as a senior annotator.
>
>     When an inter-annotator disagreement arises, the annotator in the second round reannotates the sentence and submits the results of two rounds to the senior annotator.
>
>     There are two cases of inter-annotator disagreements.
>
>     The first case is that the annotator from the first or second round makes an incorrect annotation.
>
>     e.g. 当他回到车车间时，已经有了明显的变化。(When he returned to the ga-, garage, had changed markedly.)
>
>     The sentence has two conversion problems including Disfluency and Missing Words. The annotator in the first round only annotates disfluency correctly but he/she does not catch the grammatical errors. The annotator in the second round annotates two conversion problems. Then the senior annotator will choose the correct one.
>
>     The second case is that the sentence can be corrected in multiple ways, which is common in grammatical errors.
>
>     e.g.
>
>     Source sentence: 如果人们连续看上四五个小时的电视节目，就会感到十分疲劳。(If people watch TV programs for four or five hours in a row, will feel very tired.)
>
>     Target sentence 1: 人们如果连续看上四五个小时的电视节目，就会感到十分疲劳。(People in case of watching TV programs for four or five hours in a row will feel very tired.)
>
>     Target sentence 2: 如果人们连续看上四五个小时的电视节目，他们就会感到十分疲劳。(If people watch TV programs for four or five hours in a row, they will feel very tired.)
>
>     This sentence lacks a subject and has two solutions. First, consider the conversion as Incorrect Word Order, the written language is target sentence 1. Second, consider the conversion as Missing Word, the written language is target sentence 2. In this case, the senior annotator chooses the solution with a smaller edit distance. If the edit distances are the same, the senior annotator chooses the first-round solution.
>
> **R2:** Thank you for this comment. We indeed demonstrate the value of the dataset created. We use CS2W to fine-tune two models, BART, CPT, and the results show that the fine-tuned CPT-large achieves the best results on all metrics and substantially outperforms the advanced LLM under zero-shot and few-shot (5-shot) settings, which indicates that even in the era of LLMs, CS2W still plays an important role in the Chinese Spoken-to-Written Style Conversion task.
>
> **A2:** Thank you for this comment. We indeed demonstrate the value of the dataset created. We use CS2W to fine-tune two models, BART, CPT, and the results show that the fine-tuned CPT-large achieves the best results on all metrics and substantially outperforms the advanced LLM under zero-shot and few-shot (5-shot) settings, which indicates that even in the era of LLMs, CS2W still plays an important role in the Chinese Spoken-to-Written Style Conversion task.
>
> Regarding the prompts used for zero- and few-shot setting, we tried two different prompts on BLOOM-7B under the zero-shot and 5-shot setting.
>
> Prompt1: "下面有一个口语到书面语风格转换任务, 请把口语修改为书面语: \\n\\n口语：\\n {源句} \\n书面语: \\n"
> (Here's a spoken- to written-language style conversion task, please rewrite the spoken language into the written language: \\n\\n Spoken: {source sentence} \\n Written: \\n).
>
> Prompt2: "下面有一个语法纠错任务, 请把错误的文本修改为正确的文本: \\n\\n错误文本: \\n{源句} \\n 正确文本: \\n"
> (Here's a grammatical error correction task, please correct the wrong text into the right text: \\n\\n wrong text: {source sentence} \\n correct text: \\n)
>
> Prompt2 outperforms Prompt1 on all metrics. We speculate that the model is not capable of understanding the definition of the spoken-to-written language conversion task. However, grammatical error correction is a widely-used task and the four conversion problems, except for colloquial words, can be regarded as either simple or complex grammatical errors, thus making it work well. Hence we adopt prompt2 for the rest of our experiments.
>
> Under the zero-shot setting, the prompt is “下面有一个语法纠错任务，请把错误的文本修改为正确的文本：\\n\\n错误文本：\\n {源句} \\n正确文本：\\n” (Here's a grammatical error correction task, please correct the wrong text into the right text: \\n\\n wrong text: {source sentence} \\n\\n correct text:).
>
> We expect LLMs to be able to correct all types of conversion problems. Therefore, we need to ensure the diversity of demonstrations given to LLMs. Considering that there are four types of conversion problems in CS2W, we add demonstrations of different types in turn as we have more demonstrations. The demonstrations of each conversion problem are randomly selected from the validation set, which are different from the input sentence in the test set and contain only one conversion problem. In addition, CS2W is dominated by disfluency (87.4%) as the primary conversion problem, so we select two demonstrations which are R-type and Filter word respectively under the 5-shot setting. In section 5.4, the prompt used in 5-shot experiments is “下面有一个语法纠错任务，请把错误的文本修改为正确的文本：\\n\\n错误文本：我不不太喜欢听那种歌曲；\\n正确文本：我不太喜欢听那种歌曲。\\n\\n错误文本：你呃喝酒喝的比较多了；\\n正确文本：你喝酒喝的比较多了。\\n\\n错误文本：我爸会糖醋丸子；\\n正确文本：我爸会炸糖醋丸子。\\n\\n错误文本：我想给游戏氪金；\\n正确文本：我想给游戏充钱。\\n\\n错误文本：这件事情我无可奉报；\\n正确文本：这件事情我无可奉告。\\n\\n错误文本：{源句}\\n正确文本：\\n” (Here's a grammatical error correction task, please correct the wrong text into the right text: \\n\\n wrong text: I don’t don’t like this type of music;\\n correct text: I don’t like this type of music. \\n\\n wrong text: You, uh, have been drinking a lot;\\n correct text: You have been drinking a lot.\\n\\n wrong text: My dad can sweet and sour dumplings;\\n correct text: My dad can make sweet and sour dumplings.\\n\\n wrong text:  I'd like to load up the game;\\n correct text: I'd like to top up the game.\\n\\n wrong text: I have nothing to report in this matter;\\n correct text: I have nothing to say in this matter. \\n\\n wrong text: {source sentence}\\n\\n correct text: ).
>
> In section 5.7, the prompt used for zero-shot to 5-shot is the same as the prompt above.
>
> **Question for authors:**
>
> **Q1:** A new sub-type of disfluency is described in section 4.4. Please quantify how common this type of disfluency is.
>
> **A1:** In CS2W, there are 17 sentences containing this new sub-type of disfluency. Although the number of sentences with this new type is small, it has been demonstrated that LLMs can learn to follow specific response formats from only a handful of examples in the training data without any reinforcement learning or human preference modeling [5]. Therefore, LLMs may be able to learn the feature of the new sub-type of disfluency when CS2W is used in SFT. Moreover, under the few-shot setting, the number of instances with this sub-type of disfluency is also enough to prompt LLMs. When CS2W is used for evaluation, the number is sufficient too.
>
> **Q2:** Do I understand correctly that you do not use the originally created training, development and test sets (described on lines 323f) in your experiments (because you use a different split described on lines 400f )? If so. what is the benefit of the original splits?
>
> **A2:** The experimental section (400f) shows the initial data division strategy. However, given that the dataset is relatively small, such a division according to conversion types does not make sense. Thus, the conventional randomized division is used (323f). We apologize for the confusion caused and we will correct this in the next version.
>
> **Q3:** In Figure 4, the English translation of the example is confusing: In the first line, you translate 不单 as "is not only and 不单独 as "is not". However, in the following lines, both versions are translated as "is not only". Can you explain this inconsistency? Also, you do not translate 吃. How about, e.g, "Bamboo is not only a food for pandas, but it can also be made into a lot of furniture."?
>
> **A3:** Many thanks for this. In Figure 4, "不单" should be translated as "is not only" while "不单独" should be translated as "is not individually". And your translation is better than ours. We’ll correct these in the new version.
>
> **Q4:** Could you please show/include the prompts you used for zero-shot and few-shot inference?
>
> **A4:** Please refer to our answer to the “Reason to reject”.
>
> **Q5:** Are there similar datasets in other languages than EN, CH, JA and VI, or did you focus on purpose on this selection of languages?
>
> **A5:** Thank you for your question.
>
> Publicly available spoken-to-written conversion datasets are scarce. And as of the time we completed this work, the only related dataset is Japanese S2W [1], which is mentioned in the section of related work (Line 219). Similar to ours,
> S2W is extracted from transcribed texts of spontaneous conversations. Chinese and Japanese have different linguistic conventions, thus making the type of conversions different. Japanese S2W contains conversion types: notation correction, postpositional particle expressions restoration, disfluency, and simplification. After the submission of our paper, a parallel corpus of formal-informal sentences in Persian [2] has been posted to arXiv on Aug. 10, 2023. In this corpus, informal sentences are spoken language or in a written style but used in informal conversations and social media. The data for this corpus comes from social networking sites, books, and movie scripts, while our data source is transcribed text from spontaneous conversations. It is demonstrated that disfluencies are more prevalent in spontaneous speech [3]. However, in the Informal-Formal Persian Corpus, the authors define only two types of transformations, lexical and syntactic, without focusing on disfluencies. Besides the above two datasets, there is no other spoken-to-written conversion dataset.
>
> Given that disfluency and grammatical errors dominate CS2W as the primary conversion problem, we introduce datasets for disfluency detection and grammatical error correction, respectively, in the section of related work. In the field of disfluency detection, Switchboard and PhoDisfluency are the only publicly available datasets. In addition, there are two in-house Chinese datasets [6][7], which are not publicly available. In the field of Grammatical Error Correction (GEC), there is a significant increase in open-source datasets. In addition to the Chinese and English datasets mentioned in Table 2, there are also Japanese and Spanish GEC datasets [8][9]. The GEC datasets we list in Table 2 are the more popular ones.
>
> In conclusion, for spoken-to-written style conversion and disfluency, we list all open-source datasets and the language was not chosen intentionally. For GEC, we list several most popular datasets.
>
> **Q6:** On line 198, you say that CS2W offers two annotation paradigms. Could you clarify what you mean here?
>
> **A6:** Thank you for your question. Because the dataset for the spoken-to-written style conversion task is relatively small, we refer to the annotation paradigm of many grammar error correction datasets and disfluency detection datasets for our annotation [10].
>
> There are mainly two types of annotation paradigms for constructing GEC data, i.e., error-coding and direct rewriting.
>
> The error-coding paradigm requires annotators to explicitly mark the erroneous span in the original sentence, then choose its error type, and finally make corrections. However, it is demonstrated that annotators would pay less attention to the fluency of the resulting reference when there are too many categories to consider.
>
> Instead, the direct rewriting paradigm requires annotators to directly rewrite the whole sentence, as long as the resulting sentence does not change the original meaning and is grammatically correct and fluent.
>
> In CS2W, we not only annotate the type, range of the conversion (the error-coding paradigm), but also manually rewrite the corresponding written language (the direct rewriting paradigm). That is, compared to other datasets that adopt one annotation paradigm in Table 2, CS2W adopts two annotation paradigms. We’ll make this clearer in the new version.
>
> **Q7:** Figure 3 b): Unfortunately, I don't understand this figure. Is it the number of spans per samples? (If so, I think you can remove the colors and the labels, as this seems to be redundant.)
>
> **A7:** Thank you for bringing this issue to our attention. During the annotation process, we have noticed that a sentence often has multiple conversion problems.
>
> e.g.
>
> Source sentence: 他的儿子呀是上一届奥运会的冠军嘛，并且当年世界锦标赛夺得金牌。(His son, you know, was the last Olympic champion, uh, and won the gold medal the World Championships that year.)
>
> Target sentence: 他的儿子是上一届奥运会的冠军，并且在当年世界锦标赛夺得金牌。(His son was the last Olympic champion and won the gold medal at the World Championships that year.)
>
> There are two types of conversion problems in the source sentence: Disfluency and Missing word. And there are two disfluencies including “you know” and “uh”. So, the number of conversion problem types for this sentence is 2 and the number of conversion problems is 3. And Figure 3 b)counts the number of conversion problems on a single sentence. Additionally, the colors and labels are indeed redundant, which we will correct in the next version.
>
> **Typos Grammar Style And Presentation Improvements:**
>
> **I1:** In Table 3, "The milk is a little bit very hot." Isn't this an example of an R-type disfluency instead of a redundant word? And The phrase "Incorrect Word Order" is outside the cell.
>
> **A1:** Many thanks for this. "The milk is a little bit very hot." is an example of an R-type Disfluency and we will change it with an example of redundant words in the next version.
>
> **I2:** Figure 1: In the caption, you have the abbreviation FP=Filled Pause. However, this does not appear in the example. Or should FW be FP?
>
> **A2:** Many thanks for this. In the caption of Figure 3, the abbreviation “FP=Filled” should be “FW=Filter Word”. We will change it in the next version.
>
> **I3:** Can you give an example of collocation errors (155)?
>
> **A3:** Collocation errors include subject-predicate collocation errors, predicate-object collocation errors, subject-object collocation errors, and modifier collocation errors.
>
> e.g.
>
> Source Sentence: 女足的队员就是一个球，把球踢好就是她们的任务。(The women's soccer player is a ball, and it's their job to play it well.)
>
> Target Sentence: 女足的队员就是一个整体，把球踢好就是她们的任务。(The women's soccer players are a team, and it's their job to play football well.)
>
> This is an example of a subject-object collocation error. The player cannot be a ball. So it would be more appropriate to say “the players are a team”.
>
> **I4:** What do you mean by "doesn't require annotators" (178)?
>
> **A4:** Thank you for this feedback. JFLEG adopts the rewriting paradigm while FCE and AESW adopt the error coding paradigm. The error coding paradigm requires annotation under the minimal edit distance principle [11], which always selects a reference with fewer edits when correcting errors. In contrast, the rewriting paradigm only requires that the reference be consistent with the meaning of the source sentence. Therefore, annotators are not required to perform minimal edit distance principle. We are sorry for this confusion. We’ll make it more clear in the next version.
>
> **I5:** This sentence is incorrectly constructed and confusing, please check it (277f).
>
> **A5:** The correct expression should be "CS2W is a spoken-written style conversion dataset, and the conversion problems are categorized according to Chinese linguistic characteristics. " We will correct it in the next version.
>
> **I6:** This sentence is unclear to me (311f).
>
> **A6:** In the second round of annotations, The first question (296) is to check the accuracy of the first round of annotations under the error coding paradigm (we refer to it as the annotation accuracy). The second question (298) is to check the accuracy of the first round of annotations under the rewriting paradigm (we call it as the meaning-preserving accuracy). The third question (301) is to check the consistency of the first round of annotations for both paradigms. The direct rewriting annotation requires annotators to rewrite the whole sentence, which is more flexible than the error coding paradigm. When two rounds of annotations diverge under the error coding paradigm, there is also a high probability that the results of the rewritten sentences will differ even more. So the meaning-preserving accuracy is further affected when the annotation accuracy is low. We are sorry that the expression making you feel confused, we will make this clearer in the next version.
>
> **I7:** "with disfluency/grammatical errors …" (366f): This sentence is unclear to me, isn't it a repetition of the main sentence ("Among the mixed types …")?
>
> **A7:** Thank you for this question. The meaning of the sentence (366f) is that "disfluencies are so common that many sentences with grammatical errors or colloquial words also contain disfluencies.". We are sorry that the expression making you feel confused, we will make it clearer in the next version.
>
> **I8:** Please explain what the P/R/F scores are (452).
>
> **A8:** Thank you for this question. P is precision, R is recall and F is F0.5 score, which are calculated with the MaxMatch method widely used in GEC [12]. Firstly, the gold sentence and the model’s output are word-segmented using PKUNLP word segmentation (WS) tool [13], and then we calculate the number of maximal matches according to the WS results. The P, R, F0.5 measure the different rates between the set of the model’s output edits {e1,...,e2} and the set of gold sentence edits {g1,...,g2} for all sentences:
>
> $$
> P = \frac{\sum_{i=1}^n|e_i\cap g_i|}{\sum_{i=1}^n|e_i|}
> $$
>
> $$
> R = \frac{\sum_{i=1}^n|e_i\cap g_i|}{\sum_{i=1}^n|g_i|}
> $$
>
> $$
> F_{0.5} = \frac{1.25\*P\*R}{0.25\*P+R}
> $$
>
> Where we define the intersection between $e_i$ and $g_i$ as:
>
> $$
> e_i \cap p_i = \lbrace e \in e_i |\exists g \in g_i(match(e,g)) \rbrace
> $$
>
> Other typos, grammatical and presentation issues will be fixed in the next version following your suggestion. Thank you very much.
>
> [1] Mana Ihori, Akihiko Takashima, and Ryo Masumura. 2020. Parallel corpus for japanese spoken-to-written style conversion. In Proceedings of The 12th Language Resources and Evaluation Conference, LREC 2020, Marseille, France, May 11-16, 2020, pages 6346–6353. European Language Resources Association.
>
> [2] Vahide Tajalli, Fateme Kalantari, and Mehrnoush Shamsfard. 2023. Developing an informal-formal persian corpus. CoRR, abs/2308.05336
>
> [3] Elizabeth Ellen Shriberg. 1994. Preliminaries to a theory of speech disfluencies. Ph.D. thesis, Citeseer.
>
> [4] Rohit Kundu, Preethi Jyothi, and Pushpak Bhattacharyya. 2022. Survey: Exploring disfluencies for speech-to-speech machine translation.
>
> [5] Chunting Zhou, Pengfei Liu, Puxin Xu, Srini Iyer, Jiao Sun, Yuning Mao, Xuezhe Ma, Avia Efrat, Ping Yu, Lili Yu, Susan Zhang, Gargi Ghosh, Mike Lewis, Luke Zettlemoyer, and Omer Levy. 2023. LIMA: less is more for alignment. CoRR, abs/2305.11206.
>
> [6] Shaolei Wang, Wanxiang Che, and Ting Liu. 2016. A neural attention model for disfluency detection. In COLING 2016, 26th International Conference on Computational Linguistics, Proceedings of the Conference: Technical Papers, December 11-16, 2016, Osaka, Japan, pages 278–287. ACL
>
> [7] Qianqian Dong, Feng Wang, Zhen Yang, Wei Chen, Shuang Xu, and Bo Xu. 2019. Adapting translation models for transcript disfluency detection. In The Thirty-Third AAAI Conference on Artificial Intelligence, AAAI 2019, The Thirty-First Innovative Applications of Artificial Intelligence Conference, IAAI 2019, The Ninth AAAI Symposium on Educational Advances in Artificial Intelligence, EAAI 2019, Honolulu, Hawaii, USA, January 27 - February 1, 2019, pages 6351–6358. AAAI Press.
>
> [8] Daisuke Suzuki, Yujin Takahashi, Ikumi Yamashita, Taichi Aida, Tosho Hirasawa, Michitaka Nakatsuji, Masato Mita, and Mamoru Komachi. 2022. Construction of a quality estimation dataset for automatic evaluation of japanese grammatical error correction. In Proceedings of the Thirteenth Language Resources and Evaluation Conference, LREC 2022, Marseille, France, 20-25 June 2022, pages 5565–5572. European Language Resources Association.
>
> [9] Antonios Anastasopoulos, Alison Lui, Toan Q. Nguyen, and David Chiang. 2019. Neural machine translation of text from non-native speakers. In Proceedings of the 2019 Conference of the North American Chapter of the Association for Computational Linguistics: Human Language Technologies, NAACL-HLT 2019, Minneapolis, MN, USA, June 2-7, 2019, Volume 1 (Long and Short Papers), pages 3070–3080. Association for Computational Linguistics.
>
> [10] Yue Zhang, Zhenghua Li, Zuyi Bao, Jiacheng Li, Bo Zhang, Chen Li, Fei Huang, and Min Zhang. 2022. Mucgec: a multi-reference multi-source evaluation dataset for chinese grammatical error correction. In Proceedings of the 2022 Conference of the North249
> American Chapter of the Association for Computational Linguistics: Human Language Technologies, NAACL 2022, Seattle, WA, United States, July 10-15, 2022, pages 3118–3130. Association for Computational Linguistics.
>
> [11] Ryo Nagata and Keisuke Sakaguchi. 2016. Phrase structure annotation and parsing for learner english. In Proceedings of the 54th Annual Meeting of the Association for Computational Linguistics, ACL 2016, August 7-12, 2016, Berlin, Germany, Volume 1: Long Papers. The Association for Computer Linguistics.
>
> [12] Daniel Dahlmeier and Hwee Tou Ng. 2012. Better evaluation for grammatical error correction. In Human Language Technologies: Conference of the North American Chapter of the Association of Computational Linguistics, Proceedings, June 3-8, 2012, Montréal, Canada, pages 568–572. The Association for Computational Linguistics.
>
> [13] Ruixuan Luo, Jingjing Xu, Yi Zhang, Xuancheng Ren, and Xu Sun. 2019. PKUSEG: A toolkit for multi-domain chinese word segmentation. CoRR, abs/1906.11455.

---

### Official Review · Reviewer_7Dmy · 2023-08-07

**Soundness:** 4

**Excitement:**

4: Strong: This paper deepens the understanding of some phenomenon or lowers the barriers to an existing research direction.

**Paper Topic And Main Contributions:**

Main contributions can be summarized as:

1. A new curated dataset CS2W, a Chinese Spoken-to-Written style conversion dataset comprising 7,237 spoken sentences extracted from transcribed conversational texts with four types of conversion problems.
2. Thorough analysis of the dataset, and benchmark evaluation of the on spoken-to-written language conversion with five SoTA baseline models. The analysis led to the discovery of a new type of disfluency.
3. Annotation guidelines and methodology behind the dataset preparation are shared with satisfactory detail.

**Reasons To Accept:**

1. First Chinese language spoken to written style conversion dataset with more than 7000 examples focusing on 4 conversion problems, viz., disfluency, ASR transcription error, grammatical error and colloquial words. These conversion problems are of practical importance, and have significant impact on downstream tasks based on ASR transcripts.
2. Good presentation of background work, and well-motivated research that explain the need for a dataset like CS2SW comparing to existing datasets in English, Chinese and Japanese.
3. Well presented annotation methodology as well as sound analysis of the dataset statistics and characteristics.
4. Benchmark model performance comparing five baselines trained with different pretraining methods, viz, encoder-decoder (seq2seq), decoder only, and decoder only with instruction tuning & RLHF.

**Reasons To Reject:**

1. The downstream task performance study is not convincing of general gains on any downstream task. It's hard to tell if the improvements only show on the specific setting of translation shown in the paper.
2. The dataset is dominated by disfluency (87.4%) as the primary conversion problem. It's not clear if this is expected, or an artifact of CS2W. Comparison of such stats with other datasets would be useful, if possible.

**Reproducibility:**

4: Could mostly reproduce the results, but there may be some variation because of sample variance or minor variations in their interpretation of the protocol or method.

**Reviewer Confidence:**

3: Pretty sure, but there's a chance I missed something. Although I have a good feel for this area in general, I did not carefully check the paper's details, e.g., the math, experimental design, or novelty.

---

> ### Author Rebuttal · Authors · 2023-08-29
>
> Thank you very much for your insightful comments and suggestions.
>
> **Q1:** The downstream task performance is not convincing of general gains on any downstream task. It's hard to tell if the improvements only show on the specific setting of translation.
>
> **A1:** Thank you for your question. For the machine translation tasks, we randomly selected 100 source sentences in the test set and manually translated their corresponding gold written sentences into English as references. Then we use the OPUS Chinese to English translation model to translate the source sentences and the output from the fine-tuning model (CPT-base, CPT-large) into English, and calculate BLEU.
>
> In order to demonstrate the enhancement of the spoken-to-written style conversion on the machine translation task, we increase the number of test cases. We randomly selected 200 new source sentences in addition to the 100 sentences already selected in section 5.6 and repeat the experiments. The results are shown in below table.
>
> |                 | BLEU-1 | BLEU-2 | BLEU-3 |
> | :-------------: | :----: | :----: | :----: |
> | Source sentence | 0.7672 | 0.7272 | 0.6565 |
> |    CPT-base     | 0.8293 | 0.7773 | 0.7011 |
> |    CPT-large    | 0.8520 | 0.8064 | 0.7354 |
>
> The results show that that spoken-to-written language con version can still significantly enhance the performance of machine translation tasks when the test cases are increased to 300. BLEU-4 can increase 0.0789 with CPT-large.
>
> **Q2:** Disfluency is the primary conversion type in CS2W, it is not clear if this is expected, or an artifact of CS2W. Comparison of such stats with other datasets would be useful, if possible.
>
> **A2:** Thank you for your question.
> In the field of spoken-to-written style conversion, besides CS2W, publicly available datasets include Japanese S2W [1] , which is mentioned in related work section (Line 219). Similar to our dataset, S2W is extracted from transcribed texts of spontaneous conversations. Chinese and Japanese have different linguistic conventions, thus making the type of conversions different. S2W contains conversion types: notation correction, postpositional particle expressions restoration, disfluency, and simplification. Unfortunately, the authors do not calculate the percentage of each conversion type although disfluency is also considered.
>
> In addition to S2W, a parallel corpus of formal-informal sentences in Persian [2] has recently been posted on arXiv (on Aug. 10, 2023). In this corpus, informal sentences are spoken language or in a written style but used in informal conversations and social media. Raw data for this corpus comes from social networking sites, books, and movie scripts, while our data source is transcribed text from spontaneous conversations. It is demonstrated that disfluencies are more prevalent in spontaneous speech [3]. However, in the Informal-Formal Persian Corpus, the authors define only two types of transformations, lexical and syntactic, without focusing on disfluencies.
>
> In the field of disfluency detection, Switchboard is the popular dataset, which is an English ASR telephone spontaneous conversations dataset, labeled with laughter, pause, disfluency, etc. The sentences with disfluency comprise for 5.9% of all sentences. However, as a spoken-to-written style conversion dataset, each sentence of CS2W has conversion problems, while the vast majority of sentences in the Switchboard are normal sentences that have no conversion problems. Therefore, the two statistics cannot be directly compared.
>
> In the field of dialog and Q&A, there are dialog and Q&A datasets annotated with disfluency [4][5]. However, they generally only annotate disfluency without considering other problems. This also reflects the fact that disfluency is a major problem in spontaneous speech.
>
> In addition, there are several large spoken transcription datasets similar to Switchboard  annotated with pauses, laughter but without disfluency.
>
> In summary, unfortunately, there are no other spoken-to-written style conversion dataset or spoken transcription datasets that have provided the proportion of sentences with disfluency. Our work is the first spoken-to-written style conversion dataset in Chinese, and we are the only one that has explored the conversion problem in spoken Chinese and computed the proportion of each conversion. Disfluency is a common problem in spontaneous speech, which is why disfluency detection has always been widely studied. Therefore, it is also intuitive to believe that disfluency dominates in spoken language that needs to be converted to written language.
>
> [1] Mana Ihori, Akihiko Takashima, and Ryo Masumura. 2020. Parallel corpus for japanese spoken-to-written style conversion. In Proceedings of The 12th Language Resources and Evaluation Conference, LREC 2020, Marseille, France, May 11-16, 2020, pages 6346–6353. European Language Resources Association.
>
> [2] Vahide Tajalli, Fateme Kalantari, and Mehrnoush Shamsfard. 2023. Developing an informal-formal persian corpus. CoRR, abs/2308.05336
>
> [3] Elizabeth Ellen Shriberg. 1994. Preliminaries to a theory of speech disfluencies. Ph.D. thesis, Citeseer.
>
> [4] Julian Hough, Ye Tian, Laura E. de Ruiter, Simon Betz, Spyros Kousidis, David Schlangen, and Jonathan Ginzburg. 2016. DUEL: A multi-lingual multimodal dialogue corpus for disfluency, exclamations and laughter. In Proceedings of the Tenth International Conference on Language Resources and Evaluation LREC 2016, Portorož, Slovenia, May 23-28, 2016. European Language Resources Association (ELRA).
>
> [5] Aditya Gupta, Jiacheng Xu, Shyam Upadhyay, Diyi Yang, and Manaal Faruqui. 2021. Disfl-qa: A benchmark dataset for understanding disfluencies in question answering. In Findings of the Association for Computational Linguistics: ACL/IJCNLP 2021, Online Event, August 1-6, 2021, volume ACL/IJCNLP2021 of Findings of ACL, pages 3309–3319. Association for Computational Linguistics.

---

### Meta-Review · Area_Chair_rF3v · 2023-09-18

**Recommendation:** 5

**Metareview:**

This paper describes the problem of taking transcribed Chinese text and performs a style transformation to remove disfluencies and grammatical errors, presenting an output that is more consistent with written style. To solve this problem they have prepared an open-source (non-commercial) corpus of annotated data based on an existing dataset, they have fine-tuned several existing models to establish baselines for this dataset, and have demonstrated the utility of the dataset in addressing a downstream machine translation task.

Main contributions can be summarized as:

A new curated dataset CS2W, a Chinese Spoken-to-Written style conversion dataset comprising 7,237 spoken sentences extracted from transcribed conversational texts with four types of conversion problems.
Thorough analysis of the dataset, and benchmark evaluation of the on spoken-to-written language conversion with five SoTA baseline models. The analysis led to the discovery of a new type of disfluency.
Annotation guidelines and methodology behind the dataset preparation are shared with satisfactory detail.

Reasons To Accept:
- This paper sets a standard for addressing multiple types of errors in spoken transcription through one model, with an adequate-sized corpus for future academic work.The paper appears to be free of major methodological errors, demonstrates the utility of multiple large language models against the task, and follows up by demonstrating a substantial improvement on a downstream task.

Reasons To Reject:
- There are some errors in visualization, some typos, and some missing citations, but none of these rise to the level of being reasons to reject this paper.

---

### Decision · Program_Chairs · 2023-10-07

**Decision:**

Accept-Main

**Comment:**

This paper describes the problem of taking transcribed Chinese text and performs a style transformation to remove disfluencies and grammatical errors, presenting an output that is more consistent with written style. To solve this problem they have prepared an open-source (non-commercial) corpus of annotated data based on an existing dataset, they have fine-tuned several existing models to establish baselines for this dataset, and have demonstrated the utility of the dataset in addressing a downstream machine translation task.

Main contributions can be summarized as:

A new curated dataset CS2W, a Chinese Spoken-to-Written style conversion dataset comprising 7,237 spoken sentences extracted from transcribed conversational texts with four types of conversion problems.
Thorough analysis of the dataset, and benchmark evaluation of the on spoken-to-written language conversion with five SoTA baseline models. The analysis led to the discovery of a new type of disfluency.
Annotation guidelines and methodology behind the dataset preparation are shared with satisfactory detail.

Reasons To Accept:
- This paper sets a standard for addressing multiple types of errors in spoken transcription through one model, with an adequate-sized corpus for future academic work.The paper appears to be free of major methodological errors, demonstrates the utility of multiple large language models against the task, and follows up by demonstrating a substantial improvement on a downstream task.

Reasons To Reject:
- There are some errors in visualization, some typos, and some missing citations, but none of these rise to the level of being reasons to reject this paper.